# *miR-34a* is a microRNA safeguard for *Citrobacter*-induced inflammatory colon oncogenesis

Lihua Wang[1,2,3,4], Ergang Wang[3,4], Yi Wang[3,4,5], Robert Mines[3,4], Kun Xiang[3,4], Zhiguo Sun[3,4], Gaiting Zhou[4], Kai-Yuan Chen[3,4], Nikolai Rakhilin[3,6], Shanshan Chao[1,2], Gaoqi Ye[1,2], Zhenzhen Wu[1,2], Huiwen Yan[1,2], Hong Shen[5], Jeffrey Everitt[7], Pengcheng Bu[1,2]*, Xiling Shen[3,4,6]*

[1]Key Laboratory of RNA Biology, Key Laboratory of Protein and Peptide Pharmaceutical, CAS Center for Excellence in Biomacromolecules, Institute of Biophysics, Chinese Academy of Sciences, Beijing, China; [2]University of Chinese Academy of Sciences, Beijing, China; [3]Center for Genomics and Computational Biology, Duke University, Durham, United States; [4]Department of Biomedical Engineering, Duke University, Durham, United States; [5]Affiliated Hospital of Nanjing University of TCM, Nanjing, China; [6]School of Electrical and Computer Engineering, Cornell University, New york, United States; [7]Department of Pathology, Animal Pathology Core, Duke University, Durham, United States

*For correspondence:
bupc@ibp.ac.cn (PB);
xs37@duke.edu (XS)

Competing interests: The authors declare that no competing interests exist.

**Abstract** Inflammation often induces regeneration to repair the tissue damage. However, chronic inflammation can transform temporary hyperplasia into a fertile ground for tumorigenesis. Here, we demonstrate that the microRNA *miR-34a* acts as a central safeguard to protect the inflammatory stem cell niche and reparative regeneration. Although playing little role in regular homeostasis, *miR-34a* deficiency leads to colon tumorigenesis after *Citrobacter rodentium* infection. *miR-34a* targets both immune and epithelial cells to restrain inflammation-induced stem cell proliferation. *miR-34a* targets Interleukin six receptor (IL-6R) and Interleukin 23 receptor (IL-23R) to suppress T helper 17 (Th17) cell differentiation and expansion, targets chemokine CCL22 to hinder Th17 cell recruitment to the colon epithelium, and targets an orphan receptor Interleukin 17 receptor D (IL-17RD) to inhibit IL-17-induced stem cell proliferation. Our study highlights the importance of microRNAs in protecting the stem cell niche during inflammation despite their lack of function in regular tissue homeostasis.
DOI: https://doi.org/10.7554/eLife.39479.001

## Introduction

The colon epithelium is constantly regenerated by stem cells residing at the bottoms of the intestinal crypts (*Humphries and Wright, 2008*). Infection of pathogenic bacteria in the colon can disrupt the normal gut microbiome and cause chronic inflammation, which has been linked to diseases including as inflammatory bowel disease (IBD) and recognized as a significant risk factor for colorectal cancer (CRC) development (*Gagnière et al., 2016*; *Wang and Karin, 2015*; *Collins et al., 2011*). It has been estimated that chronic inflammation and persistent infections contribute to a significant portion of human cancers, especially CRC (*Wang and Karin, 2015*; *Zur Hausen, 2009*).

Inflammation plays a dual role in tissue homeostasis. On one hand, inflammation is associated with damage to the tissue; on the other hand, it triggers stem cell proliferation and reparative regeneration (*Karin and Clevers, 2016*). Events of damage and inflammation have been associated with regenerative signaling pathways such as Wnt to increase the number of stem cells and cause

regeneration and hyperplasia in intestinal and colonic epithelia (*Ashton et al., 2010*; *Miyoshi et al., 2012*).

Inflammation triggers intestinal and colonic epithelial reparative regeneration via inflammatory cytokines, including TNF-a, IL-6, IL-17, and IL-22. These cytokines upregulate downstream pathways such as MAPK, JAK-STAT3, and NF-κB, which control processes including cell proliferation and differentiation (*Karin and Clevers, 2016*; *Chen et al., 2003*; *Taniguchi et al., 2015*; *Sugimoto et al., 2008*; *Song et al., 2011*). Deficiency in IL-22 or IL-17 Receptor E (IL-17RE) led to enhanced mucosal damage after infection by pathogenic bacteria such as *Citrobacter rodentium* (*Song et al., 2011*; *Zheng et al., 2008*).

On the other hand, chronic inflammation causes excessive regeneration, and the resulting hyperplasia could eventually lead to cancer. TNF-$\alpha$ is associated with CRC progression (*Al Obeed et al., 2014*; *Zins et al., 2007*), and blocking TNF-$\alpha$ reduces the likelihood of colorectal carcinogenesis associated with chronic colitis (*Popivanova et al., 2008*). IL-17 have also been shown to promote colitis-associated early colorectal carcinogenesis (*Grivennikov et al., 2009*; *Wang et al., 2014*), and IL-22 stimulates stem cell growth after injury and promotes CRC stemness (*Lindemans et al., 2015*; *Kryczek et al., 2014*). Infiltration of T helper 1 (Th1) cells in CRC tumor specimens is associated with prolonged disease-free survival. However, infiltration of T helper 17 (Th17) cells, which secrete IL-17 and IL-22, is predictive of poor prognosis for CRC patients (*Tosolini et al., 2011*).

The microRNA *miR-34a* is an important tumor suppressor that targets pro-growth genes (*He et al., 2007*; *Chang et al., 2007*), and its mimics are among the first microRNA mimics to reach clinical trial for cancer therapy (*Bouchie, 2013*; *Bader, 2012*). *miR-34a* also limits self-renewal of cancer stem cells (*Bu et al., 2013*; *Bu et al., 2016*; *Liu et al., 2011*). *miR-34a* expression is often silenced in various cancer types (*Lodygin et al., 2008*; *Kong et al., 2012*; *Corney et al., 2010*), and methylation of the *miR-34a* promoter is correlated with CRC progression (*Siemens et al., 2013*; *Wang et al., 2016*). Nevertheless, *miR-34a* deficiency alone does not increase susceptibility to spontaneous tumorigenesis (*Cheng et al., 2014*; *Jiang and Hermeking, 2017*; *Concepcion et al., 2012*), raising many questions about the role of *miR-34a* in tissue homeostasis.

In this study, we demonstrate that *miR-34a* acts as safeguard to protect the stem cell niche during inflammation-induced reparative regeneration. *miR-34a* deficiency led to colon tumorigenesis after *C. rodentium* infection, where Th17 cell infiltration and epithelial stem cell proliferation were observed. During the pro-inflammatory response, *miR-34a* suppressed Th17 cell differentiation and expansion by targeting IL-23R, Th17 cell recruitment to the colon epithelium by targeting CCL22, and IL-17 induced stem cell proliferation by targeting IL-17RD. Loss of *miR-34a* results in a reparative regeneration process that goes awry.

## Results

### *C. rodentium* infection promotes colon carcinogenesis and stem cell enrichment in *miR-34a-/-* mice

Microbial dysbiosis causes chronic inflammation associated with CRC (*Sobhani et al., 2013*; *Candela et al., 2011*; *Plottel and Blaser, 2011*; *Tjalsma et al., 2012*). *C. Rodentium* is a mouse mucosal pathogen that shares pathogenic mechanisms and 67% of its genes with enteropathogenic *Escherichia coli* (EPEC) and enterohaemorrhagic *E. coli* (EHEC), which are two clinically important human gastrointestinal pathogens (*Schauer and Falkow, 1993a*; *Schauer and Falkow, 1993b*; *Papapietro et al., 2013*; *Borenshtein et al., 2008*; *Borenshtein et al., 2007*; *Gibson et al., 2008*). *C. Rodentium* has been used as a model to study mucosal immunology, including intestinal inflammatory responses during bacteria-induced colitis and colon tumorigenesis (*Collins et al., 2014*; *Chandrakesan et al., 2014*; *Higgins et al., 1999*). *C. rodentium* infection increases the number of colonic adenomas in ApcMin mice but does not cause adenoma formation in wild-type mice (*Newman et al., 2001*).

After *C. rodentium* ($2 \times 10^9$ CFU) infection, both wild-type and *miR-34a-/-* mice developed diarrhea and weight loss within 2 weeks. Elongation of crypts and loss of goblet cells were observed (*Figure 1A*). Histopathological changes of pre-neoplasia and neoplasia were limited to the *miR-34a-/-* genotype and were first noted at the four-month time point. Microscopic sections from wild type control mice were free of dysplastic and neoplastic changes at four-month and six-month time

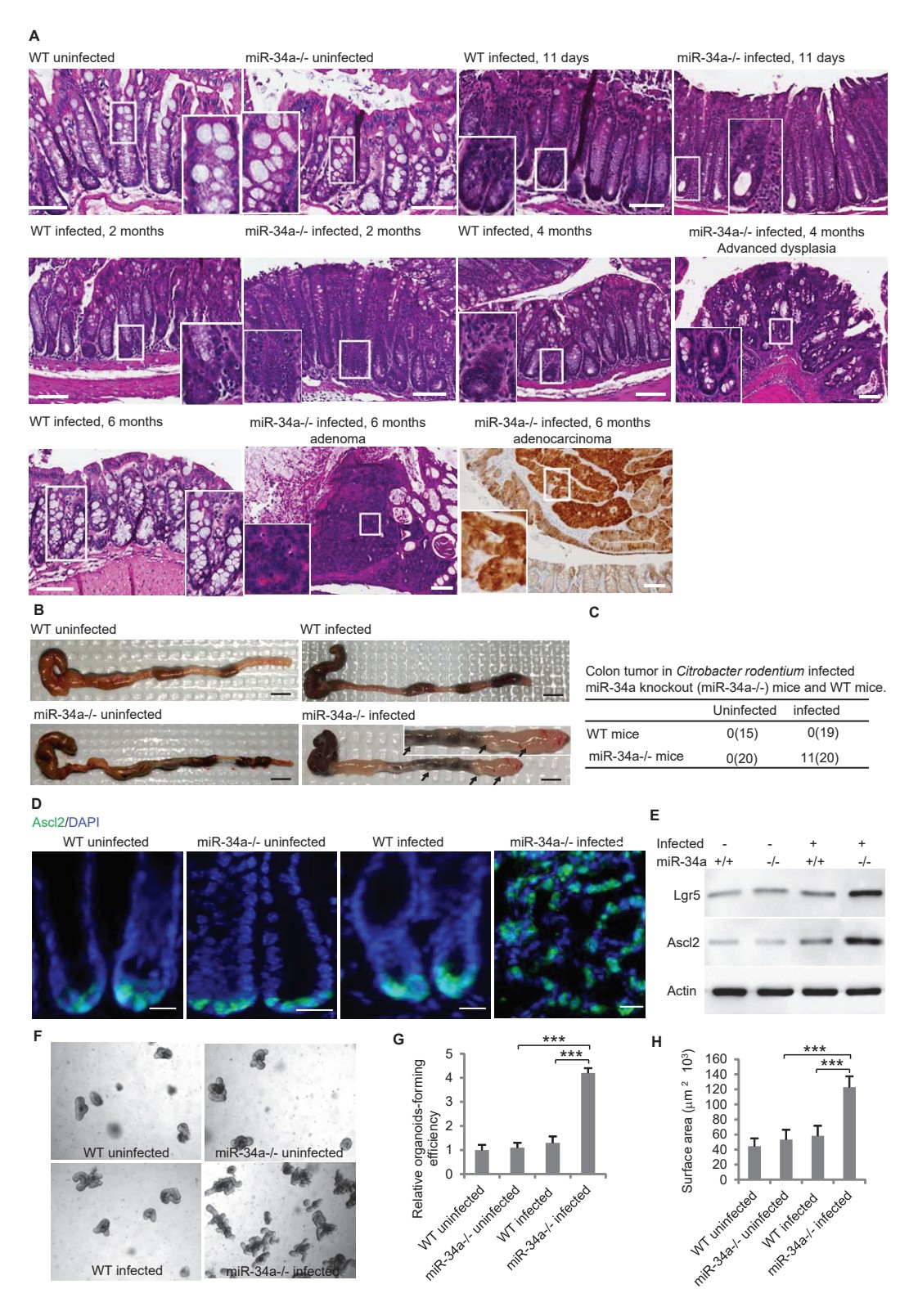

**Figure 1.** *C.rodentium* infection induces colonic tumor formation and stem cell enrichment in *miR-34a-/-* mice. (**A**) Representative H and E staining of colon tissues from infected and uninfected wildtype (WT) and *miR-34a-/-* mice. Scale bar, 100 μm. (**B**) Representative images of mouse colons uninfected or infected with *C. rodentium*. The arrows indicate the visible colon tumors. $2 \times 10^9$ CFU *C. rodentium* were used to infect the mice orally. Six months after the infection, the mice were euthanized and the colons were imaged. Scale bar, 5 mm. (**C**) Frequencies of colonic tumor formation in

*Figure 1 continued on next page*

*Figure 1 continued*

infected and uninfected mice. (D) Immunofluorescence of Ascl2 showing enriched colon stem cells in *miR-34a-/-* colon tumors. Scale bar, 40 µm. (E) Western blot of Ascl2 and Lgr5 showing enriched colon stem cells in miR-34a-/- colon tumors. (F–H) Representative colon organoids images (F) and quantification showing organoid-forming efficiency (H) and organoid sizes. Colon organoids were cultured from uninfected mice and *C. rodentium* infected mice after 2 months. 1000 organoid cells from each condition were reseeded to examine organoid-forming efficiency and organoids growth.

DOI: https://doi.org/10.7554/eLife.39479.002

The following figure supplements are available for figure 1:

**Figure supplement 1.** *C.rodentium* infection induces colonic tumor formation in miR-34a-/- mice.

DOI: https://doi.org/10.7554/eLife.39479.003

**Figure supplement 2.** No detectable APC mutation in tumors from infected *miR-34a-/-* mice.

DOI: https://doi.org/10.7554/eLife.39479.004

points following infection (*Boivin et al., 2003*). In *miR-34a-/-* mice, no dysplasia or early neoplasia was present at a two-month time point (0/4), whereas at four months half the animals (2/4) had dysplastic change microscopically. At the six-month time point, 11 out of 20 miR 34a-/- mice had microscopic changes ranging from dysplasia (2/20), to adenoma (7/20), to adenocarcinoma (2/20) (*Figure 1A–C*, *Figure 1—figure supplement 1A*). All tumors in this model were relatively well differentiated. One animal with a colonic adenocarcinoma in the section of distal colon also had a squamous cell carcinoma of the rectum. Dysplastic and neoplastic changes were characterized by strong intracytoplasmic β-catenin staining and occasional cells with nuclear staining (*Figure 1A*). The earliest dysplastic changes are noted in deep reaches of crypts that are within inflamed ulcerated colonic mucosa in several of the sections where the diffuse inflammation of the *C. rodentium* has subsided and focal long-standing inflammation has set up due to ulceration of the surface. (*Figure 1—figure supplement 1A*). No APC mutations were detected in the tumors from the infected *miR-34a-/-* mice (*Figure 1—figure supplement 2*). No liver or lung metastasis was found in the *C. rodentium*-infected mice (*Figure 1—figure supplement 1B,C*).

The colon stem cells, marked by the Wnt signaling enhancers Lgr5 and Ascl2 (*Schuijers et al., 2015*), are usually confined at the base of the crypt in wild-type and *miR-34a-/-* mice but became enriched in *C. rodentium*-induced colon tumors in *miR-34a-/-* mice (*Figure 1D*). Enrichment of Lgr5 and Ascl2 expression in the colon tumors of infected *miR-34a-/-* mice was further confirmed by western blot (*Figure 1E*). Colonic epithelial cells from infected *miR-34a-/-* mice (2 months post infection) had significantly higher organoid-forming efficiency and growth rates that the other groups (*Sato et al., 2011a*; *Sato et al., 2011b*) (*Figure 1F–H*).

## Th17 cells are enriched in number and in close proximity to stem cells in *miR-34a-/-* colon tumors

CD4 +T helper (Th) cells are known to infiltrate and accumulate in the inflammatory environment, which can either promote or suppress tissue malignancy (*Terzić et al., 2010*; *Kim and Bae, 2016*). We isolated CD4 +Th cells from the colon epithelium of *C. rodentium*-infected wild-type and *miR-34a-/-* mice and analyzed the relative abundance of Th1, Th2, Th17, and Treg subpopulations according to their associated expression of IFN-γ, IL-4, IL-17, and FoxP3, respectively. IFN-γ, IL-4, and FoxP3 levels were similar between wild-type and *miR-34a-/-* tissue, but IL-17 was significantly upregulated in *miR-34a-/-* tissue (*Figure 2A*). *miR-34a* deletion increased the number of CD4 +IL17 +Th17 in the colon 2 and 6 months post *C. rodentium* infection (*Figure 2B,C*). Immunofluorescence suggested that many of the enriched CD4 +IL-17 + Th17 cells were in proximity to Ascl2 +colon stem cells (*Figure 2D,E*).

In the infected colon, miR-34a deletion did not significantly increase the number or IL-17 expression level of lineage(CD3e/Ly-6G/Ly-6C/CD11b/CD45R/B220/TER-119)-/CD117+/CD45 +cells, which contain a subset of ILC3 cells that may express IL-17 (*Dong, 2008*) (*Figure 2—figure supplement 1A,B*). Nevertheless, more markers will be needed to distinguish ILC3 and its subtypes specifically. Similarly, the macrophage and neutrophil populations were only slightly affected (*Figure 2—figure supplement 1C,D*).

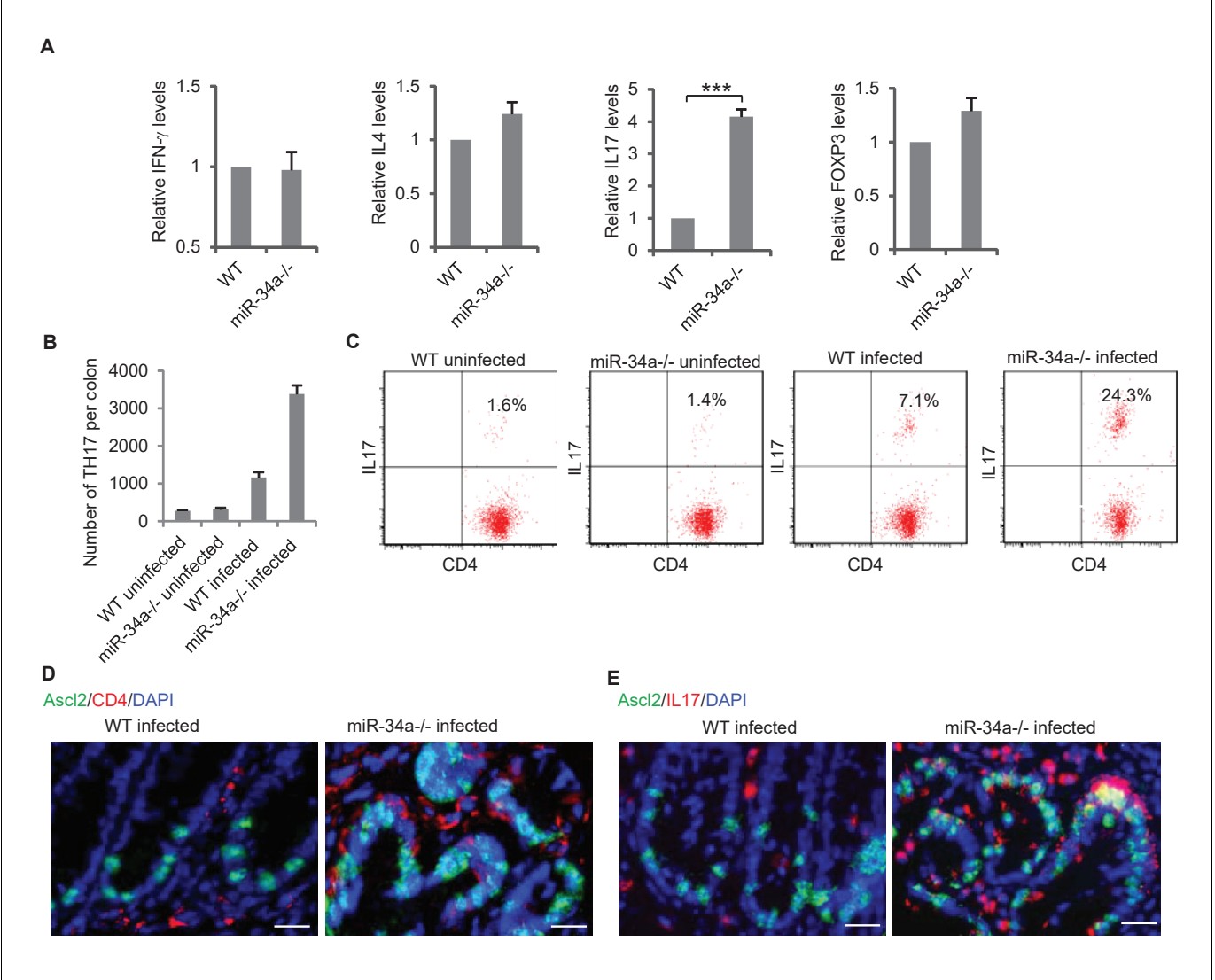

**Figure 2.** *C.rodentium* infection enhances Th17 cell infiltration in *miR-34a-/-* colonic tumors. (**A**) RT-qPCR showing relative expression of the CD4+ T lymphocyte genes associated with Th1 (*IFN-γ*), Th2 (*IL-4*), Th17 (IL-17), and Treg (FOXP3) cells in the colons from *C. rodentium* infected wild-type and *miR-34a-/-* mice. (**B**) FACS analysis showing Th17 cells (CD4+/IL-17+) numbers in infected and uninfected mice colon at month 2. (**C**) FACS analyses of CD4 +T cells from each infected and uninfected WT and *miR-34a-/-* mice colon at month 6. The percentages of Th17 cells (CD4+/IL-17+) are marked. (**D and E**) Immunofluorescence of CD4 (**D**) and IL-17 (**E**) in infected colons showing enhanced Th17 cells infiltrating in *miR-34a-/-* colonic tumors. Scale bar, 40 μm. Error bars denote s.d. of triplicates. ***p<0.001. p-value was calculated based on Student's t-test.

DOI: https://doi.org/10.7554/eLife.39479.005

The following figure supplement is available for figure 2:

**Figure supplement 1.** Presence of innate immune cells in *C. rodentium*-infected wild-type and *miR-34a-/-* mice.

DOI: https://doi.org/10.7554/eLife.39479.006

## *miR-34a* suppresses Th17 differentiation and expansion by targeting IL6R and IL-23R

We then aimed to understand how *miR-34a* deletion led to accumulation of Th17 cells in the *C. rodentium*-induced colon tumors. IL-6 is critical for initiating the differentiation of native CD4 +T cells into Th17 cells, and IL-23 is essential for the final step of Th17 cell differentiation, its proliferation, and IL-17 expression (*Dong, 2008*; *Acosta-Rodriguez et al., 2007*; *Toussirot, 2012*). We found that protein levels of IL-6R and IL-23R, the receptors for IL-6 and IL-23, were upregulated in CD4 +T cells isolated from the *miR-34a-/-* colon compared to the wild-type control (*Figure 3A*, *Figure 3—*

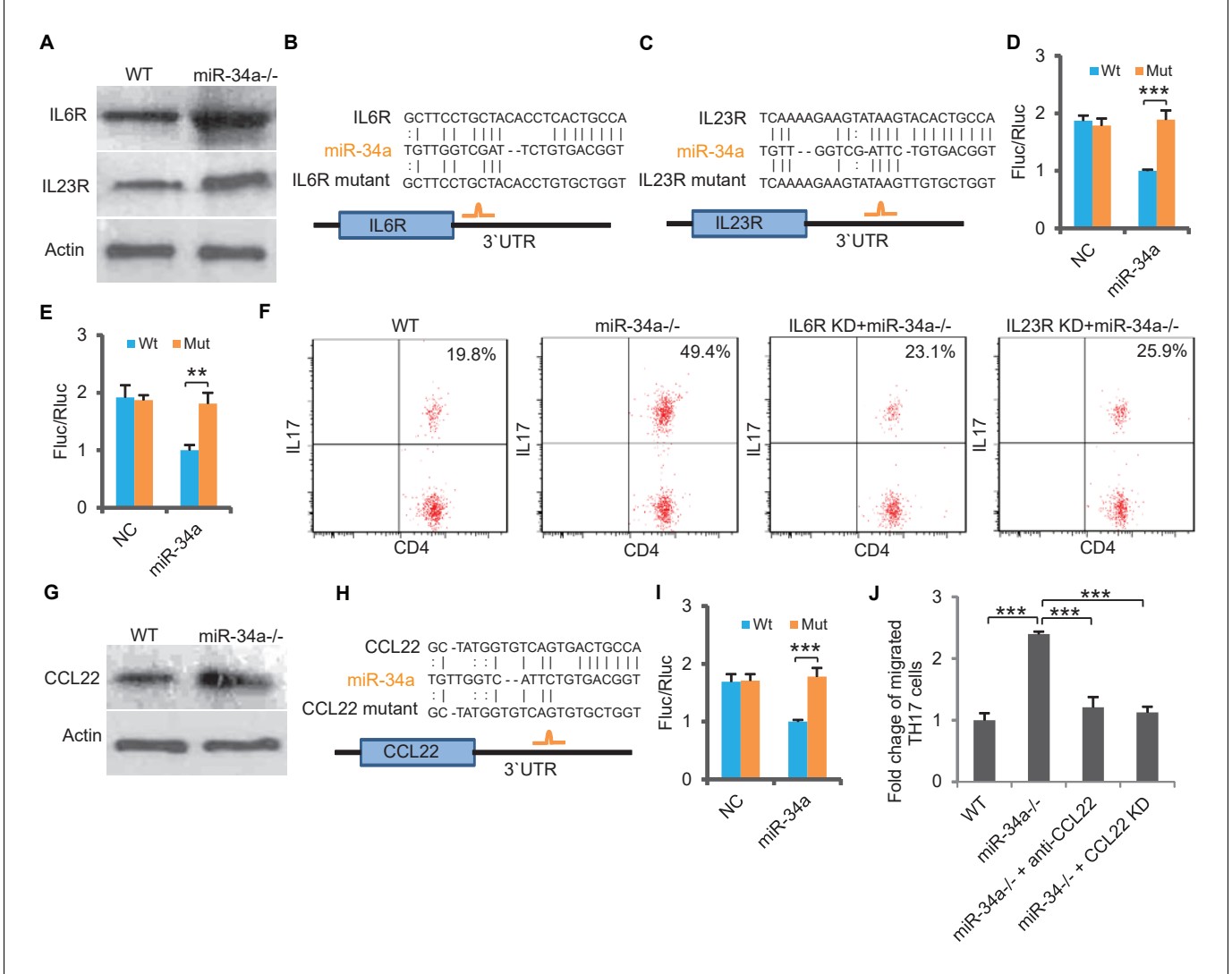

**Figure 3.** *miR-34a* targets IL-6R, IL-23R and CCL22. (**A**) Western blot showing IL-6R and IL-23R expression levels in CD4 +T cells isolated from *C. rodentium* infected colon of wild-type and *miR-34a-/-* mice. (**B and C**) Schematic representation of mouse *IL-6r* (**B**) and *IL-23r* (**C**) 3'UTRs containing the putative *miR-34a* binding sites. (**D and E**) Luciferase reporter assays confirming the *miR-34a* binding sites. 3'UTRs of mouse *IL-6r* (**D**) and *IL-23r* (**E**) containing wild-type (Wt) or mutated (Mut) putative *miR-34a* binding sites were cloned into the 3'UTR of firefly luciferase (Fluc). Ectopic *miR-34a* expression in CT26 cells downregulated luciferase in Wt cells, but not in Mut cells. Fluc signals were normalized by a simultaneously delivered Renillar luciferase (Rluc) expression plasmid. (**F**) FACS showing knockdown of *IL-6r* or *IL-23r* in CD4 +T cells largely offsets the effect of *miR-34a* loss on Th17 cell differentiation. (**G**) Western blot showing increase of CCL22 expression in *miR-34a-/-* colon crypts. (**H**) Schematic representation of *miR-34a* binding site on the mouse *CCL22* 3'UTR. (**I**) Luciferase reporter assays confirming the miR-34a binding sites in mouse *CCL22* 3'UTR. (**J**) Chemotaxis assay showing knockdown of *CCL22* in colon tumor organoid cells or neutralization of CCL22 with anti-CCL22 antibody suppresses Th17 cell migration to colon tumor organoid conditioned medium. Error bars denote s.d. of triplicates. **p<0.01; ***p<0.001. p-value was calculated based on Student's t-test.

DOI: https://doi.org/10.7554/eLife.39479.007

The following figure supplements are available for figure 3:

**Figure supplement 1.** *miR-34a* loss upregulates target genes in uninfected mice.
DOI: https://doi.org/10.7554/eLife.39479.008
**Figure supplement 2.** Validation of gene knockdown efficiency.
DOI: https://doi.org/10.7554/eLife.39479.009
**Figure supplement 3.** Loss of *miR-34a* enhances CCL22 expression in colon epithelium.
DOI: https://doi.org/10.7554/eLife.39479.010
**Figure supplement 4.** Global gene expression in colon epithelial and CD4 +T cells from wildtype vs. *miR-34a-/-* mice.
DOI: https://doi.org/10.7554/eLife.39479.011

*figure supplement 1A,B*). The RNA22 algorithm identified putative *miR-34a* binding sites in the *IL-6r* and *IL-23r* 3'UTRs (*Figure 3B,C*), which were then confirmed by the luciferase reporter assay (*Figure 3D,E*).

To evaluate the *miR-34a*/IL-6R and *miR-34a*/IL-23R axes for Th17 cell differentiation, we performed an in vitro Th17 differentiation assay (*Esplugues et al., 2011*) using CD4 +T cells isolated from the wild-type and *miR-34a-/-* mice. Loss of *miR-34a* significantly enhanced CD4 +T cell differentiation into Th17 cells, which was largely abrogated by knockdown of either *IL-6r* or *IL-23r* (*Figure 3F*, *Figure 3—figure supplement 2*). Therefore, *miR-34a* suppresses Th17 cell differentiation by targeting IL-6R and IL-23R.

## *miR-34a* suppresses Th17 recruitment by targeting CCL22

Th17 cells express chemokine receptors CCR6 and CCR4 (*Antonic et al., 2013*), and the CCR6/CCL20 and CCR4/CCL22 axes play important roles in Th17 cell migration (*Borenshtein et al., 2008*). Loss of *miR-34a* did not affect *CCR6* or *CCR4* expression in CD4 +T cells (*Figure 3—figure supplement 3A*). However, *CCL22* expression in the colon epithelium was significantly upregulated in the *miR-34a-/-* mice compared to the wild-type, while *CCL20* expression remained unchanged (*Figure 3G*, *Figure 3—figure supplement 3B*). It has been previously reported that a TGF-β-*miR-34a*-CCL22 axis promotes venous metastases of HBV-positive liver cancer (*Yang et al., 2012*). Our western blot and luciferase reporter assays confirmed that *miR-34a* suppressed *CCL22* by directly targeting its 3'UTR in colon epithelial cells (*Figure 3G–I*).

Conditioned medium collected from *miR-34a-/-* colon tumor organoids enhanced the migration of *in vito*-differentiated Th17 cells in comparison to medium from the wild-type colon organoids (*Figure 3J*). The addition of anti-CCL22 neutralizing antibody in the medium or knockdown of *CCL22* in *miR-34a-/-* colon tumor organoids reduced Th17 migration back to the wild-type level (*Figure 3J*, *Figure 3—figure supplement 2*). Therefore, *miR-34a* suppresses recruitment of Th17 cells by targeting CCL22 production in colon epithelial cells.

## Th17 cells promote colon organoid growth via IL-17

We then tested whether Th17 cells, which were enriched by loss of *miR-34a* and were located in proximity to Ascl2 +colon stem cells (*Figure 2B–D*), regulated colon epithelial cell proliferation. Mouse CD4 +T cells were induced to differentiate into Th17 cells and co-cultured with colon organoids. The presence of Th17 cells significantly increased the organoid sizes. Accordingly, the addition of anti-IL-17 neutralizing antibody suppressed this growth, suggesting that the growth effect was caused by Th17-secreted IL-17 (*Figure 4A*). In the absence of Th17 cells, recombinant IL-17 in the medium increased organoid growth (*Figure 4B,C*) and also upregulated Lgr5 and Ascl2 (stem cell marker) expression (*Figure 4D*).

## IL-17 activates STAT3 signaling via NF-κB

It has been reported that STAT3 activation is involved in Enterotoxigenic *E. coli*-induced colon carcinogenesis in ApcMin mice (*Kopan and Ilagan, 2009*). We treated mouse colon organoids with recombinant IL-17 and measured STAT3 phosphorylation by western blot. IL-17 activated STAT3, which was abrogated by the STAT3 inhibitor, Stattic (*Figure 4E*, *Figure 4—figure supplement 1A*). Inhibition of STAT3 by Stattic impaired colon organoid growth (*Figure 4F–G*, *Figure 4—figure supplement 1B*). IL-17 activates NF-κB in addition to STAT3 (*Figure 4H*). Treatment of colon organoids with an NF-κB inhibitor, BAY 11–7082, suppressed IL-17-induced STAT3 phosphorylation and organoids growth (*Figure 4H–J*). Hence, IL-17 seems to activate STAT3 through NF-κB, which promotes colon organoids growth.

## *miR-34a* targets IL-17RD to suppress stem cell proliferation

The IL-17 receptor, IL-17RA, is essential for IL-17-mediated signaling (*Bility et al., 2014*). IL-17RD, an orphan IL-17 receptor, has been reported to interact with IL-17RA to mediate IL-17 signaling (*Li et al., 2009*). RT-qPCR showed that the *IL-17ra* transcript levels were similar between *C. Rodentium*-induced *miR-34a-/-* colon tumors and the wild-type colon, whereas the *IL-17rd* transcript levels were significantly increased in *miR-34a-/-* colon tumors (*Figure 5A*). Western blot confirmed that the

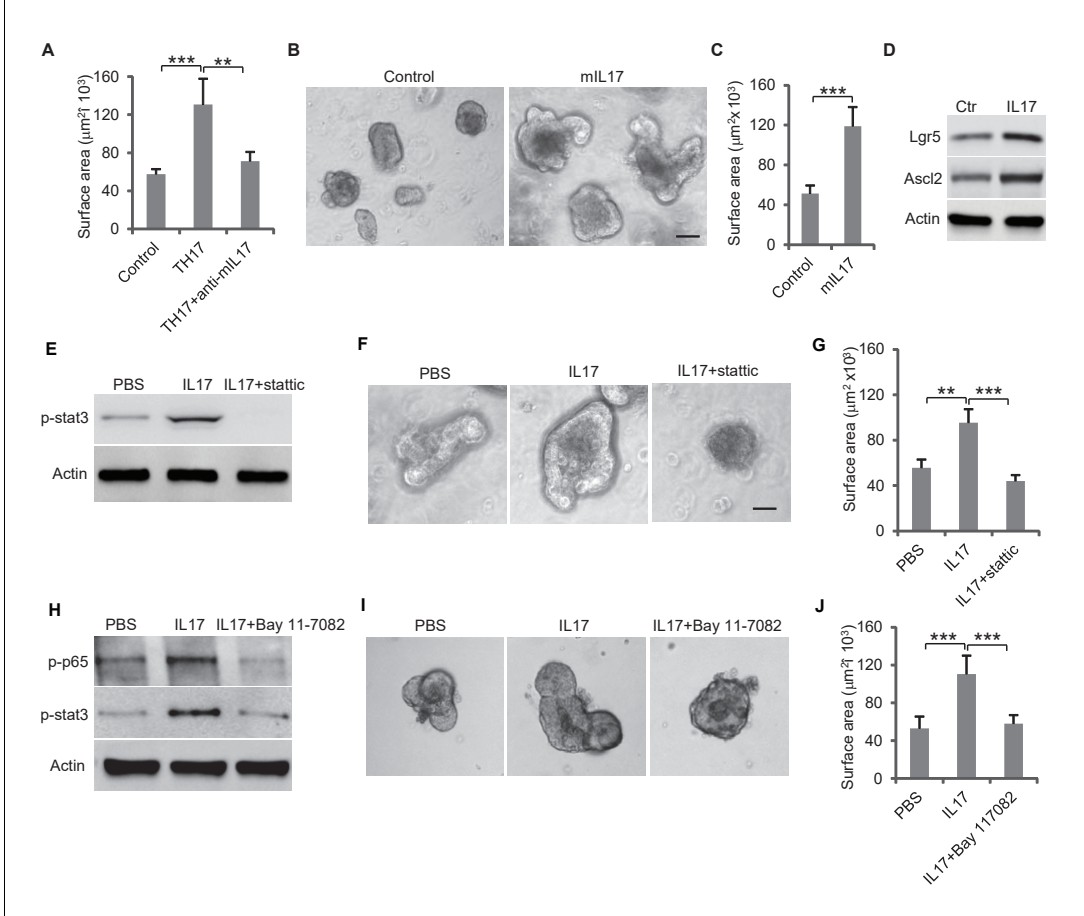

**Figure 4.** Th17 cells enhance colon organoid growth through IL-17. (**A**) Th17 cells enhance colon organoid growth in co-culture. When co-cultured with Th17 cells, colon organoids grow faster with bigger surface area. Anti-IL-17 antibody abrogates Th17 promotion of colon organoids growth. (**B and C**) Recombinant mouse IL-17 enhances mouse organoids growth as shown by representative mouse colon organoids images (**B**) and quantitative organoid area (**C**). (**D**) Western blot showing that mouse IL-17 increases the expression of colon stem cell markers, Ascl2 and Lgr5, in mouse colon organoids. (**E**) Western blot of phospho-stat3 with IL-17 (20 ng) and STAT3 inhibitor, stattic (20 μM). (**F and G**) Representative organoid images (**F**) and quantification of organoid area (**G**) with IL-17 and stattic. (**H**) Western blot of phospho-stat3 and phospho-p65 with IL-17 (20 ng) and an NF-κB inhibitor, BAY 11–7802 (5 μM). (**I and J**) Representative organoid images (**I**) and quantificaiton of organoid area (**J**) with IL-17 and BAY 11–7802. Error bars denote s.d. of triplicates. **p<0.01; ***p<0.001. p-value was calculated based on Student's t-test. .

DOI: https://doi.org/10.7554/eLife.39479.012

The following figure supplement is available for figure 4:

**Figure supplement 1.** IL-17 activates STAT3 and promotes organoid growth.
DOI: https://doi.org/10.7554/eLife.39479.013

IL-17RD protein level was upregulated in the *miR-34a-/-* colon epithelial cells and tumor (*Figure 5B*, *Figure 3—figure supplement 1D*).

RNA22 predicted a *miR-34a* binding site in the *IL-17rd* 3'UTR (*Figure 5C*). The luciferase reporter assay confirmed that *miR-34a* directly targets the 3'UTR of *IL-17rd* and suppresses IL-17RD expression (*Figure 5D*). Mutation of the endogenous *miR-34a* binding site in the *IL-17rd* 3'UTR by CRISPR/CAS9 increased IL-17RD expression in wildtype mouse organoids but not in *miR-34a-/-* organoids (*Figure 5—figure supplement 1*). Co-immunoprecipitation confirmed interaction between IL-17RA and IL-17RD in mouse colon crypts (*Figure 5E*). Knockdown of either *IL-17ra* or *IL-17ra* inhibited IL-17-mediated STAT3 activation (phosphorylation) and colon organoid growth (*Figure 5F–H*, *Figure 3—figure supplement 2*). Furthermore, colon organoid growth spurred by loss of *miR-34a* was largely offset by *IL-17rd* knockdown (*Figure 5I*).

Consistent with its safeguard role, *miR-34a* levels were higher in Lgr5-GFP + colon stem cells than in Lgr5-GFP- cells (*Figure 5—figure supplement 2A*). On the other hand, *miR-34b* and *miR-*

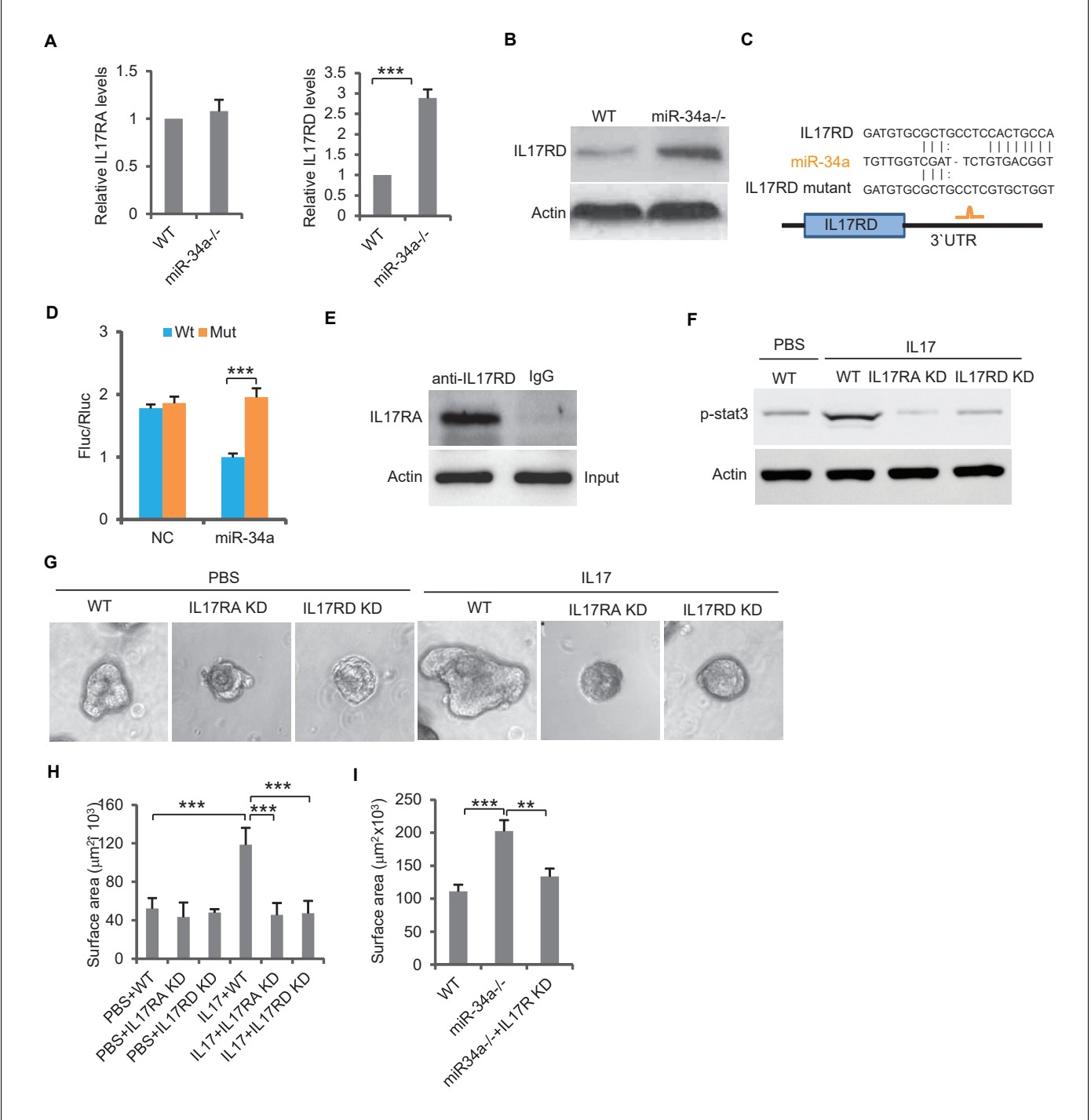

**Figure 5.** *miR-34a* targets orphan receptor IL-17RD in colon stem cells to suppresses IL-17-induced growth. (**A**) RT-qPCR showing relative expression of *IL-17ra* and *IL-17rd* in *C. rodentium* infected *miR-34a-/-* colonic tumors and wildtype controls. (**B**) Western blot showing increase of IL-17RD expression in *miR-34a-/-* colonic tumors. (**C**) Schematic representation of mouse *IL-17rd* 3'UTR and the putative miR-34a binding site. (**D**) Luciferase reporter assay confirming the *miR-34a* binding sites in mouse *IL-17rd* 3'UTR. (**E**) Immunoprecipitation showing the IL-17RA and IL-17RD complex in the colon crypt. (**F**) Western blot showing IL-17RA and IL-17RD is required for IL-17 mediated STAT3 activation. (**G and H**) *IL-17RA* and *IL-17RD* knockdown suppresses IL-17 mediated colon organoids growth as shown by representative organoids images (**G**) and quantitative organoids surface area (**H**). (**I**) IL-17RD knockdown reduces *miR-34a* deficiency-induced colon organoids growth. Error bars denote s.d. of triplicates. **p<0.01; ***p<0.001. p-value was calculated based on Student's t-test.

DOI: https://doi.org/10.7554/eLife.39479.014

The following figure supplements are available for figure 5:

*Figure 5 continued on next page*

*Figure 5 continued*

**Figure supplement 1.** Validation of *miR-34a* targeting IL-17RD using CRISPER/CAS9.

DOI: https://doi.org/10.7554/eLife.39479.015

**Figure supplement 2.** Expression of *miR-34a*, miR-34b and *miR-34c*.

DOI: https://doi.org/10.7554/eLife.39479.016

*34c*, the other two *miR-34* family members, were barely detectable in colon epithelial cells (*Figure 5—figure supplement 2B*). Furthermore, *miR-34a* expression levels in both CD4 +T cells and colon epithelial cells were upregulated during *C. rodentium*-induced inflammation (*Figure 5—figure supplement 2C,D*). This is consistent with the dual role *miR-34a* plays in both CD4 +T cells and colon epithelial cells.

## Both epithelial and immune *miR-34a* deficiencies contribute to tumorigenesis

Given that *miR-34a* regulates both the immune and epithelial cells, we assessed their individual contributions to *C. rodentium*-induced colon tumorigenesis. We generated a *miR-34a* conditional knockout mice strain *Lgr5-EGFP-CreER^{T2}/miR-34a^{flox/flox}* by crossing *miR-34a^{flox/flox}* mice with *Lgr5-EGFP-IRES-CreER^{T2}* mice (*Bu et al., 2016*). In this strain, intraperitoneal injection of Tamoxifen deletes *miR-34a* in Lgr5-EGFP + stem cells and their progeny (*Figure 6A*, *Figure 6—figure supplement 1*). The *miR-34a* conditional knockout did not affect IL6R and IL23R expression in CD4 +T cells, but increased CCL22 and IL17RD expression in the colon epithelium (*Figure 3—figure supplement 1E–H*). As in the uninfected *miR-34a-/-* mice, *Lgr5-EGFP-CreER^{T2}/miR-34a^{flox/flox}* mice did not develop colon tumors spontaneously. When infected with *C. rodentium*, 1 out of 7 mice developed colon tumors at the end of our 9 month observation period (*Figure 6B*). The number of CD4 +IL17+Th17 cells increased in the infected colons of *Lgr5-EGFP-CreER^{T2}/miR-34a^{flox/flox}* mice relative to wild-type mice (*Figure 6C*, *Figure 6—figure supplement 2*), but not to the degree of *miR-34a-/-* mice as shown in *Figure 2B*.

We then performed bone marrow transplantation by replacing the bone marrow in irradiated wildtype C57BL/6J mice with the bone marrow from *miR-34a-/-* mice, which resulted in mice with wildtype epithelial cells but *miR-34a-/-* immune cells (*Figure 6D*). The transplantation efficiency was validated to be above 90% by flow analysis (*Figure 6—figure supplement 3*). 6 months after *C. rodentium* infection, 3 out of the 12 mice with transplanted *miR-34a-/-* bone marrow developed colon tumors, whereas none of the wildtype mice developed tumors (*Figure 6E*). CD4 +IL17+Th17 cells were enriched in the infected colons of the mice with *miR-34a-/-* bone marrow transplants (*Figure 6F*). Caveats of this transplantation experiment include the potential confounding effects of radiation on the intestinal cells (e.g., LGR5 +cells are sensitive to radiation) and radiation-resistant cells, which can only be addressed by additional control groups with bone marrow transplantation from wild-type to wild-type and from *miR-34a-/-* to *miR-34a-/-*. Taken together, the combination of *miR-34a* deficiency in both epithelial and immune cells seems to elicit a stronger tumorigenic effect than *miR-34a* deficiency in epithelial or immune cells alone, consistent with the interaction between Th17 cells and colon epithelial cells.

## IL-17 neutralizing antibody abrogates *C. rodentium*-induced colon stem cell proliferation and tumorigenesis

To validate the role of IL-17 in *C. rodentium*-infection-induced colon stem cell proliferation, we infected the mice with *C. rodentium* on day 0, then intraperitoneally injected the mice with IL-17 neutralizing antibody on days 1, 5, and 10, and sacrificed them on either day 15 or day 30 to harvest the colon (*Figure 6G*). Injection of the IL-17 antibody suppressed Ascl2 +stem cells proliferation in the infected colonic crypts according to immunofluorescence (*Figure 6H*) and decreased Ascl2 and Lgr5 expression according to western blot (*Figure 6I*).

We then intraperitoneally injected *miR-34a-/-* mice with IL-17 neutralizing antibody on day 1, 5, 10 after *C. rodentium* infection, and then every 15 days up to 2 months. 7 out of 13 *miR-34a-/-* mice developed colon tumors in the isotype control group, whereas only 1 out of 12 *miR-34a-/-* mice

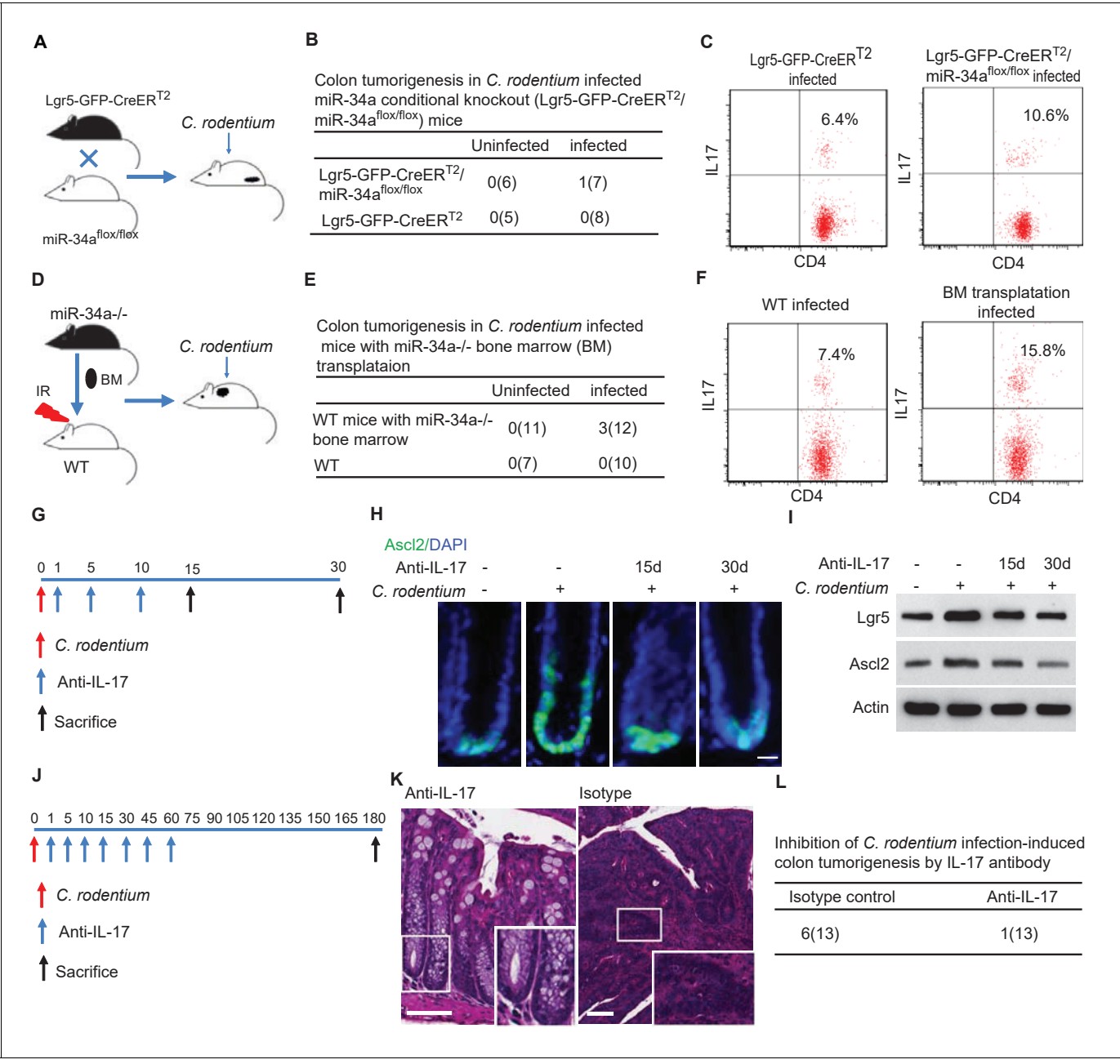

**Figure 6.** Conditional *miR-34a* knockout in LGR5-EGFP + stem cells and in bone marrow transplanted immune cells contribute to tumorigenesis and Th17 cell accumulation while IL-17 neutralizing antibody blocks stem cell proliferation and tumorigenesis. (A) Schematic of the *C. rodentium*-infected *Lgr5-EGFP-CreER^T2/miR-34a^flox/flox* mouse model. (B) Frequencies of colon tumor formation in *C. rodentium* infected and uninfected *Lgr5-EGFP-CreER^T2/miR-34a^flox/flox* mice model. (C) FACS analyses of Th17 cells (CD4+/IL-17+) in *C. rodentium* infected and uninfected *Lgr5-EGFP-CreER^T2/miR-34a^flox/flox* mice colon. (D) Schematic of the *C. rodentium*-infected *miR-34a-/-* bone marrow transplant mouse model. (E) Frequencies of colonic tumor formation in *C. rodentium* infected and uninfected *miR-34a-/-* bone marrow transplant mice. (F) FACS analyses of Th17 cells (CD4+/IL-17+) in *C. rodentium* infected and uninfected *miR-34a-/-* bone marrow transplant mice colon. (G) Schematic of the IL-17 neutralizing antibody treatment. 500 ug isotype control or IL-17 antibody were intraperitoneally injected at the indicated days. (H) Immunofluorescence of Ascl2 showing IL-17 antibody largely abrogated *C. rodentium*-infection-induced colon cancer stem cell proliferation. Scale bar, 40 μm. (I) Western blot of Ascl2 and Lgr5 showing IL-17 antibody-abrogated *C. rodentium*-infection-induced colon stem cell marker expression. (J) Schematic of the IL-17 neutralizing antibody treatment. 500 μg isotype control or IL-17 antibody were intraperitoneally injected at the indicated days. (K) Representative H and E staining of colon tissues from IL-17 antibody or isotype control treated mice. Scale bar, 50 μm. (L) Frequencies of colonic tumor formation in IL-17 antibody or isotype control treated mice.
DOI: https://doi.org/10.7554/eLife.39479.017
The following figure supplements are available for figure 6:

*Figure 6 continued*

**Figure supplement 1.** Validation of miR-34a conditional knockout mice.
DOI: https://doi.org/10.7554/eLife.39479.018
**Figure supplement 2.** Numbers of Th17 cells in infected *Lgr5-EGFP-CreER^T2^/miR-34a^flox/flox^* mouse colons.
DOI: https://doi.org/10.7554/eLife.39479.019
**Figure supplement 3.** Validation of bone marrow transplantation efficiency.
DOI: https://doi.org/10.7554/eLife.39479.020

developed colon tumors in the antibody treated group (*Figure 6J–L*). Hence IL-17 antibody treatment inhibited *C. rodentium* infection induced colon tumorigenesis.

## IL-17 and *miR-34a* levels are associated with human CRC

To investigate whether IL-17 regulates human colon epithelial growth, we grew organoid cultures from human colon tissue using an established protocol (*Sato et al., 2011a*) and added recombinant human IL-17 into the culture medium. Consistent with mouse organoids, addition of human IL-17 into the medium increased the sizes of human colon organoids (*Figure 7A,B*). RT-qPCR

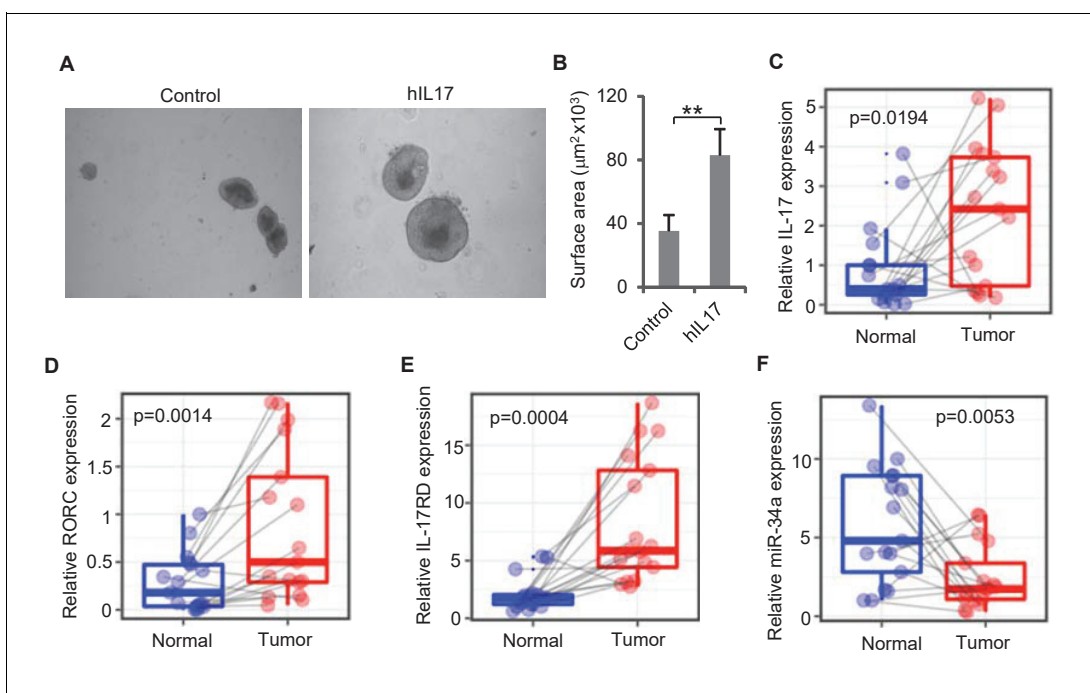

**Figure 7.** IL-17 and *miR-34a* expression in human CRC. (**A and B**) Human IL-17 enhances human colon organoids growth as shown by representative human colon organoids images (**A**) and quantitative organoids area (**B**). Error bars denote s.d. of triplicates. **p<0.01; p-value was calculated based on Student's t-test. (**C–F**) RT-qPCR of colonic tumor and normal colon tissue samples from CRC patients (*Figure 7—source data 1*) comparing *IL-17, RORC, IL-17RD*, and *miR-34a* transcript levels. Dots refer to different samples, and lines connect the paired samples. Error bars denote s.e.m. of 17 normal and cancer samples. p-values were calculated based on paired t-test.
DOI: https://doi.org/10.7554/eLife.39479.021

The following source data and figure supplement are available for figure 7:

**Source data 1.** Source data for *Figure 7*.
DOI: https://doi.org/10.7554/eLife.39479.023
**Source data 2.** Source data for *Figure 7*.
DOI: https://doi.org/10.7554/eLife.39479.024
**Figure supplement 1.** miR-34a targets human IL-6R, IL-17RD and CCL22.
DOI: https://doi.org/10.7554/eLife.39479.022

measurements of matched tumor and normal colon samples from 17 CRC patients (*Figure 7—source data 1*) suggested that the expression levels of the two Th17 cell markers *IL-17* and *RORC* as well as *miR-34a* target *IL-17rd* were higher in tumor tissues than that in normal colon tissues, while *miR-34a* expression levels were downregulated (*Figure 7C–F*).

We then validated whether *miR-34a* suppresses IL-6R, IL-23R, CCL22, and IL-17RD in human cells. We ectopically expressed *miR-34a* in Jurkat cells, a human T lymphocyte cell line, and SW480 cells, a human colon epithelial cancer cell line. Western blots indicated that *miR-34a* suppressed IL-6R expression in Jurkat cells as well as CCL22 and IL-17RD expression in SW480 cells. However, *miR-34a* did not seem to suppress IL-23R expression (*Figure 7—figure supplement 1*).

We then checked the *miR-34a* binding sequences in the 3'UTRs of the human genes using miRanda and RNA22. Luciferase reporter assays containing the 3'UTRs with wild-type or mutated binding sequences confirmed that *miR-34a* directly binds to these putative binding sites in *IL-17rd*, CCL22 and *IL-6r* 3'UTRs and regulates their expression (*Figure 7—figure supplement 1*). On the other hand, *miR-34a* did not seem to regulate IL-23R.

## Discussion

*miR-34a* is a known tumor suppressor that targets cell proliferation and apoptosis genes. In fact, RNA-seq performed on splenic CD4 +T cells and colon epithelial cells isolated from *miR-34a-/-* and wildtype mice revealed various changes in gene expression (*Figure 3—figure supplement 4*), including well-known *miR-34a* target genes such as Notch1, Snai2, BCL2, and c-Met (*Figure 8—source data 1* and *2*). However, *miR-34a* deficiency alone does not lead to tumorigenesis, suggesting that mere upregulation of these genes by *miR-34a* loss is not sufficient to cause cancer. Our study indicates that *miR-34a* acts as a safeguard for the inflammatory stem cell niche and reparative regeneration by modulating both the immune and epithelial responses to infection and inflammation (*Figure 8*). First, *miR-34a* suppresses Th17 cell differentiation and expansion by targeting IL-6R and IL-23R. Second, *miR-34a* limits Th17 cell recruitment to the epithelium by targeting CCL22. Lastly, *miR-34a* hinders IL-17-induced stem cell proliferation by targeting IL-17RD. These *miR-34a* targeting mechanisms are largely conserved between murine and human, although *miR-34a* does not seem to target IL-23R in human T cells.

Colon stem cells reside at the base of the crypt, relying on the niche to provide necessary signaling cues for self-renewal. The mesenchyme beneath the niche provides many essential factors (*Degirmenci et al., 2018*). In addition, cKit+/Reg4 +colonic crypt base secretory cells interdigitate with Lgr5 +stem cells, providing the latter with Notch ligands DLL1 and DLL4, and epidermal growth factor (*Rothenberg et al., 2012*; *Sasaki et al., 2016*). Normally, stem cells are constrained to this spatial niche and are forced to differentiate when they leave the niche. However, in human colon adenoma and carcinoma samples, Lgr5 +stem like cells are highly upregulated and are not confined to the spatial niche as in normal crypts (*Baker et al., 2015*). Similarly, we have observed this enrichment and expanded distribution of stem cells in *C. rodentium*-induced colon tumors in *miR-34a-/-* mice. Inflammatory cytokines such as IL-17 potentially provide an enlarged 'inflammatory niche' by stimulating receptors such as IL-17RD on the stem cells, enabling them to ignore the constraints of the crypt base and proliferate away from the crypt base mesenchyme and secretory cells. Interestingly, IL-17RD amplifies IL-17RA signaling in a way analogous to Lgr5 receptor amplification of Wnt signaling for self-renewal.

Non-coding RNAs occupy the majority of the mammalian genome (*Kung et al., 2013*; *Mattick and Rinn, 2015*). Evolutionarily, the percentage of genome devoted to the non-coding region is consistently associated with the complexity of the organism, rising from less than 25% in prokaryotes, 25–50% in simple eukaryotes, more than 50% in fungi, plants and animals, to approximately 98.5% in humans—which have a genome size that is three orders of magnitude larger than prokaryotes (*Mattick, 2004*). Compared to microRNA, the role of long non-coding RNA (lncRNA) in regulating tumors has just started to be appreciated (*Huarte, 2015*; *Prensner and Chinnaiyan, 2011*; *Schmitt and Chang, 2016*). In fact, lncRNA has been shown to regulate *miR-34a* in human CRC, especially in cancer stem cells (*Wang et al., 2016*). Similar to *miR-34a*, many of the lncRNAs with strong functions in tumors are largely dispensable for normal development and tissue homeostasis (*Zhang et al., 2012*; *Nakagawa et al., 2014*). It is possible that the abundance of non-coding RNAs in mammals may provide extra surveillance to protect tissue integrity during stress conditions

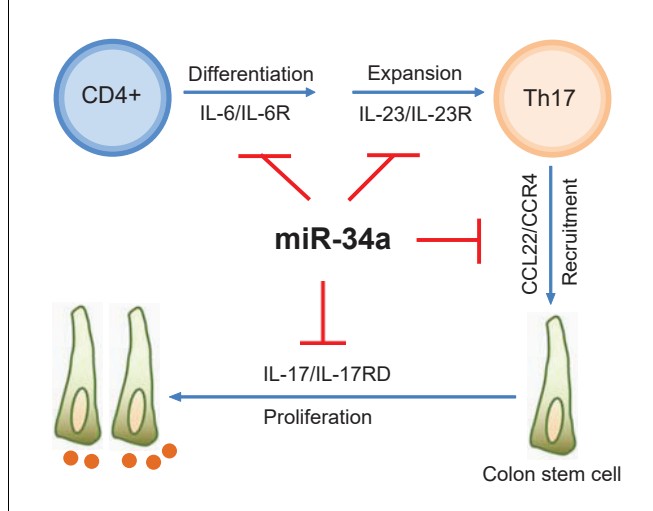

**Figure 8.** with two source data *miR-34a* regulates Th17 cell-mediated proliferation. A schematic illustration of the central role of *miR-34a* in Th17 cell-mediated colon stem cell proliferation. *miR-34a* suppresses Th17 cell differentiation and expansion by targeting IL-6R and IL-23R in immune cells. *miR-34a* further inhibits Th17 cells recruitment by targeting CCL22 in the colon epithelium. *miR-34a* also inhibits IL-17RD expression to suppress IL-17-IL-17RD/IL17-RA-mediated colon stem cell proliferation.
DOI: https://doi.org/10.7554/eLife.39479.025

The following source data is available for figure 8:

**Source data 1.** Transcriptomic profiling of epithelial cells.
DOI: https://doi.org/10.7554/eLife.39479.026
**Source data 2.** Transcriptomic profiling of CD4+ T cells.
DOI: https://doi.org/10.7554/eLife.39479.027

such as inflammation, which are often not captured by laboratory animal models raised in well-controlled circumstances.

The *miR-34a* mimic was the first microRNA mimic to reach clinical trial for cancer therapy (*Bouchie, 2013*; *Bader, 2012*). Previous studies largely focused on the role of *miR-34a* to induce cell cycle arrest, senescence, and apoptosis. This study suggests that enhancing *miR-34a* levels may have additional benefits of suppressing Th17 cells and IL-17 stimulation of cancer stem cells in the tumor microenvironment. Furthermore, inflammatory cytokines such as IL-6 can suppress *miR-34a* to increase epithelial-mesenchymal transition (EMT) (*Rokavec et al., 2014*), so boosting *miR-34a* may mitigate inflammation-induced CRC invasion and migration. It might be worth paying extra attention to CRC with Th17 cell enrichment and evaluating therapeutics effects based on CRC classification, especially on the inflammatory and stem cell-like subtypes (*Sadanandam et al., 2013*).

# Materials and methods

## Transgenic mice and bacterial infection

C57BL/6J and *B6(Cg)-Mir34atm1Lhe/J* mice (*Choi et al., 2011*) were ordered from the Jackson Laboratory. *Lgr5-EGFP-creER^{T2}/miR-34a^{flox/flox}* mice were generated as described as previously (*Bu et al., 2016*). Cre recombinase was induced by intraperitoneal injection of tamoxifen (Sigma) dissolved in sterile corn oil at a dose of 75 mg/kg before infection with *C. rodentium*. Mouse maintenance and procedures were approved by Duke University DLAR and followed the protocol (A286-15-10). *C. rodentium* strain DBS100 was purchased from ATCC and cultured according to previously described methods (*Shui et al., 2012*). $2 \times 10^9$ C.F.U *C. rodentium* were infected into 8 weeks old mice by oral gavage.

## Bone marrow transplantation

The bone marrow transplantation procedures were performed as previously described (*Alpdogan et al., 2003*). Male C57BL/6J.SJL mice (Ly5.1) with the *Ptprc*[b] leukocyte marker CD45.1/ Ly5.1 obtained from the Jackson Laboratory were used as recipients for transplantation at the age of 8–10 weeks. The recipient mice received 1000 Rad (1Gy, filter 4) whole body lethal irradiation on XRad320. After 6 hr, the irradiated mice received bone marrow cells from male C57BL/6J miR34a -/- donor mice with *Ptprc*[b] leukocyte marker CD45.2/Ly5.2. The donor femurs were collected aseptically, and the bone marrow canals were washed out with sterile media. 6 million cells per mouse were transplanted into lethally irradiated recipients via tail vein injection. Mice were housed and received sulfamethoxazole trimethoprim medicated, acidified water for 4 weeks. 6 weeks after reconstitution, blood was collected from the recipient mice, and the reconstituted blood cell lineages were analyzed by flow cytometry using CD45.1-PE (Biolegend) and CD45.2-FITC (Biolegend).

## Clinical specimen and colon organoid culture

Frozen CRC specimens and paired controls were acquired from Weill Cornell Medical College (WCMC) Colon Cancer Biobank for evaluation of Th17 cell-related gene expression. Surgically resected fresh normal human colon tissues were obtained from Duke University hospital. The study was approved by the ethical committee of Duke University hospital, Duke University, and WCMC. All samples were obtained with informed consent.

Mouse and human colon crypt isolation and organoid culturing were performed as described previously (*Sato et al., 2011a*). To investigate CCL22 regulation on Th17 cell migration and IL-17RA and IL-17RD regulation on organoids growth, lentiviral vector carrying shRNA against *CCL22*, *IL-17ra* or *IL-17rd* were purchased from Sigma and infected into organoids according to previously described protocols (*Koo et al., 2011*).

## CD4 +T cell isolation and Th17 cell differentiation

To investigate Th17 cell enrichment in *C. rodentium*-infected colons, CD4 +T cells were first isolated from mouse colon as described previously (*Weigmann et al., 2007*). Briefly, after washing with cold PBS, the mouse colon was cut into 0.5–1 cm pieces and incubated in $Ca^{2+}$ and $Mg^{2+}$ free PBS containing 0.37 mg/ml EDTA and 0.145 mg/ml DTT in a shaking incubator at 37°C for 15 min. The supernatant was decanted, and the remaining tissue was further incubated in RPMI-1640 containing 5% fetal calf serum, 20 mM HEPES, 100 U/ml each of penicillin and streptomycin, and 0.1 mg/ml collagenase dispase (Sigma) while shaking at 37°C for 90 min. After filtering through 70 μm cell strainers, the cells were collected by centrifugation, and the pellet was suspended in 35% percoll solution (Sigma). The cells were then collected by centrifugation at 2000 rpm for 20 min. CD4 +T cells were isolated using a mouse CD4 +T cell isolation kit (StemCell Technology). Then 10,000 CD4 +T cells were counted for flow cytometry. After staining for CD4 and IL-17, Th17 cells were analyzed by flow cytometry.

To evaluate the effect of IL-23R on Th17 cell differentiation, CD4 +T cells were isolated from mouse spleen as described previously (*Weigmann et al., 2007*). Briefly, the spleen was minced and squeezed through 70 μm cell strainers to obtain single cells. After collection by centrifugation, the cells were suspended into 35% percoll solution (Sigma) with heparin, followed by incubation in red cell lysis buffer (Abcam) to get rid of red blood cells. The cells were then washed, and CD4 +T cells were isolated using a mouse CD4 +T cell isolation kit (StemCell Technology). Isolated CD4 +T cells were cultured in 24-well plate coated with anti-CD3e and anti-CD28 antibodies in 1640 RPMI medium with 10% FCS and recombinant mouse IL-2 (rmIL2, 20 ng/mL) at $1 \times 10^6$/ mL according to the previous protocol (*Zhong et al., 2010*). Lentiviral vectors carrying shRNA against *IL-23r* were purchased from Sigma and infected into CD4 +T cells following previously described protocols (*Bao et al., 2006*). After selection by antibiotics, the cells were induced to differentiate into Th17 cells using the FlowCellect Mouse Th17 Differentiation Kit according to the protocol (EMD Millipore). Th17 cell differentiation efficiency was measured by flow cytometry by CD4 and IL-17 staining.

## Co-culture Th17 cells with organoids

After differentiation from CD4 +T cells, Th17 cells were co-cultured with colonic crypts at a 10:1 ratio in Matrigel. To activate and maintain Th17 cells, rmIL-2 (20 ng/ml; Pepro-tech), mIL-6 (50 ng/ml; Pepro-tech), TGF-β (10 ng/ml; Pepro-tech), mIL-23 (30 ng/ml; Pepro-tech) were added into the ENR organoids culture medium. A neutralizing monoclonal antibody against IL-17(Abcam) was used to abrogate IL-17 specific effects of Th17 cells.

## Histochemical staining

Selected colon tissues from wildtype and *miR-34a-/-* mice euthanized at 2 days, 2, 4, and 6 months after *C. rodentium* infection were fixed in 10% neutral buffered formalin, processed routinely and embedded in paraffin, sectioned at five microns, and stained with hematoxylin and eosin. Glass slides were scanned via high resolution virtual slide imaging at 40x (Aperio, Leica Biosystems) and then reviewed by a board-certified veterinary pathologist with experience in mouse tumor biology (JE) without knowledge of genotype. Lesions were scored according to established murine pathology (*Boivin et al., 2003*). Representative proliferative colonic lesions were selected for recuts and β-catenin immunohistochemistry (IHC) was performed. IHC was conducted with a rabbit monoclonal antibody against β-catenin (1:400, Abcam) after epitope retrieval. The secondary detection system was a labelled polymer-HRP anti-rabbit (Dako).

## Chemotaxis assays

The chemotaxis assay was performed as described previously (*Huang et al., 2008*). Briefly, $1 \times 10^5$ Th17 cells were applied to the upper well of the ChemoTex chambers (96-well, 5 µm pore size; NeuroProbe). Conditional medium from *miR-34a-/-* colon organoids or control organoids was added in the lower chamber. To evaluate the effect of CCL22 on Th17 migration, a neutralizing monoclonal antibody against CCL22 (R and D) was included in the conditional medium. After a 2 hr incubation, the cells in the upper wells were removed, and the migrated cells were collected by centrifugation. Migrated cells were counted using a hemocytometer.

## Immunofluorescence

Immunofluorescence was performed on paraffin-embedded colon sections. After rehydration and antigen retrieval, the sections were blocked by 2% horse serum in PBS for 2 hr at RT and incubated with anti-Ascl2 (1:200, Santa Cruz), anti-IL17 (1:200, Abcam), anti-CD4 (1:50, R and D Systems) or anti-GFP (1:500, Abcam) primary antibodies in antibody diluent buffer (DAKO) overnight at 4°C. After washing, the sections were then incubated with Rhodamine Red or Alexa fluor 488 labeled secondary antibodies (Invitrogen) for 1 hr at room temperature. After counterstaining with DAPI (Invitrogen), the slides were observed on an Axio Imager upright microscope (Zeiss).

## Flow cytometry analysis

Th17 cells were analyzed by CD4 and IL-17 staining. Briefly, single cells were fixed with 4% formaldehyde and further permeabilized by methanol. The cells were then incubated with anti-IL-17 (1:200, Abcam) and anti-CD4 (1:100, R and D Systems) antibodies, followed by incubation with APC or FITC labeled secondary antibody (Invitrogen). The samples were analyzed using a Beckman Coulter flow cytometer. The raw FACS data were analyzed with the FlowJo software.

## Quantitative real-time PCR

Total RNA was extracted from the tissue using the RNeasy mini kit (Qiagen). cDNA was synthesized from 500 ng of total RNA in 20 µl of reaction volume using the High Capacity cDNA Archive Kit (Applied Biosystems). Quantitative PCR was carried out using TaqMan assays (Applied Biosystems) to detect *miR-34a* and hIL-17, and the SYBR Green System (Applied Biosystems) for all other gene expression measurements. *miR-34a* primers were purchased from Applied Biosystems, and hIL-17 primers were purchased from Thermo Fisher. Other qPCR primers were synthesized by IDT, and the sequences are listed in *Figure 7—source data 2*. All relative gene expression measurements utilized at least three biological replicates for both the wild-type and miR34a deficient experimental groups with three technical replicates per biological replicate. The expression of each gene was defined

using the threshold cycle (Ct), and the relative expression levels were calculated using the 2-△△Ct method after normalization to the ß-actin expression level.

## Western blot

Whole cell lysate was prepared in a RIPA lysis buffer (Millipore) with protein inhibitor (Roche). Proteins were first separated by 10% SDS-PAGE and then transferred to a Hybond membrane (Amersham). The membranes were incubated with primary antibodies for anti-lgr5 (1:500, Santa Cruz), anti-ASCL2 (1:1000, Bioss), anti-IL23R (1:500, R and D Systems), anti-CCL22(1:500, R and D Systems), anti-IL17RD (1:500, R and D Systems), anti-IL17RA (1:500, R and D Systems), anti-pSTAT3 (1:1000, Cell Signaling) or anti-actin (1:2000, Cell Signaling) in 5% milk/TBST buffer (25 mM Tris pH 7.4, 150 mM NaCl, 2.5 mM KCl, 0.1% Triton-X100) overnight and then probed for 2 hr with secondary horseradish peroxidase (HRP)-conjugated anti-goat or anti-rabbit IgG (Santa Cruz). After an extensive wash with PBST, the target proteins were detected on membrane by enhanced chemiluminescence (Pierce).

Sequence mutation and gene knockdown gRNA:CAGAATGATGGCGGTGGCAG-TGG was designed for mutation of sequence complementary to *miR-34a* binding site in the mouse *IL-17rd* 3UTR. The gRNA was then vector pLentiCRISPR v2. and transfected into mouse single colon stem cells. DNA sequencing of single colonies confirmed successful deletion of *miR-34a* binding in mouse *IL-17rd* 3UTR. Mutagenesis for luciferase reporter assay was performed using QuickChange Site-Directed Mutagenesis Kit (Stratagene). shRNAs for knockdown of *IL-6R*, *IL-23R*, *IL-17RA*, *IL-17RD* and *CCL22* were purchased from Sigma. PCR were performed using primers to amply three most APC mutation regions in mouse colon cancer. Primers: 'GCCATCCCTTCACGTTAG' and 'TTCCAC TTTGGCATAAGGC' for DNA sequence contains mutation 1; Primers: 'TGACAGCACAGAATCCAG TG' and 'TACCAAGCATTGAGAG' for DNA sequence contains mutation 2 (B); Primers: 'TAGGCAC TGGACATAAGGGC' and 'GTAACTGTCAAGAATCAATGG' for DNA sequence contains mutation 3.

## Statistical analysis

Data were expressed as mean ± standard deviation (s.d.) of three biological replicates. Student T-tests were used for comparisons, with $p < 0.05$ considered significant. Patient data were expressed as mean ± standard error of the mean (s.e.m.). Paired T-tests were used for comparison of the 17 matched patient normal colon and CRC samples.

## Acknowledgements

This work was supported in part by NIH R35GM122465, NIH R01GM114254, NIH R21CA201963, NSF 1350659, NSF 1137269, NSF 1511357, NSF GRFP 1644868, National Natural Science Foundation of China (31771513), Strategic Priority Research Program of the Chinese Academy of Sciences (XDB29040000), CAS Pioneer Hundred Talents Program, Chinese Ministry of Science and Technology (2017YFA0504103), Thousand Young Talents Program of China. We have no conflict of interest.

## Additional information

### Funding

| Funder | Grant reference number | Author |
|---|---|---|
| National Natural Science Foundation of China | 31771513 | Pengcheng Bu |
| Chinese Academy of Sciences | XDB29040000 | Pengcheng Bu |
| Chinese Academy of Sciences | Pioneer Hundred Talents Program | Pengcheng Bu |
| Chinese Ministry of Science and Technology | 2017YFA0504103 | Pengcheng Bu |
| National Institute of General Medical Sciences | R35GM122465 | Xiling Shen |
| National Cancer Institute | U01CA214300 | Xiling Shen |

| National Science Foundation | 1350659 | Xiling Shen |
|---|---|---|
| National Cancer Institute | U01CA217514 | Xiling Shen |
| National Institute of General Medical Sciences | R01GM114254 | Xiling Shen |
| National Institutes of Health | R21CA201963 | Xiling Shen |
| National Science Foundation | 1137269 | Xiling Shen |
| National Science Foundation | 1511357 | Xiling Shen |
| National Science Foundation | GRFP 1644868 | Xiling Shen |

The funders had no role in study design, data collection and interpretation, or the decision to submit the work for publication.

## Author contributions
Lihua Wang, Conceptualization, Validation, Investigation, Visualization, Methodology, Writing—original draft, Conceived of the concept, Designed and performed the experiments, Co-wrote the manuscript; Ergang Wang, Validation, Investigation, Visualization, Methodology, Helped with computational analysis and other experiments; Yi Wang, Validation, Investigation, Visualization, Methodology, Helped with animal experiment and some other experiments; Robert Mines, Validation, Investigation, Visualization, Methodology, Helped with RT-qPCR and other experiments; Kun Xiang, Investigation, Visualization, Helped with western blot; Zhiguo Sun, Investigation, Methodology, Helped with organoids culture; Gaiting Zhou, Investigation, Visualization, Helped with immunostaining; Kai-Yuan Chen, Methodology, Helped with CRISP technology and data analysis; Nikolai Rakhilin, Investigation, Helped with animal experiments; Shanshan Chao, Validation, Investigation, Helped with western blot; Gaoqi Ye, Investigation, Validated RT-qPCR results; Zhenzhen Wu, Validation, Investigation, Performed some RT-qPCR; Huiwen Yan, Investigation, Performed some RT-qPCR; Hong Shen, Methodology, Validated the histology results; Jeffrey Everitt, Validation, Visualization, Analyzed and validated the histology results; Pengcheng Bu, Conceptualization, Supervision, Funding acquisition, Methodology, Project administration, Writing—review and editing, Conceived of the concept, Designed the experiments, Co-wrote the manuscript; Xiling Shen, Conceptualization, Resources, Supervision, Visualization, Project administration, Writing—review and editing, Conceived of the concept, Designed the experiments, Co-wrote the manuscript

## Author ORCIDs
Ergang Wang (ID) http://orcid.org/0000-0003-2550-3686
Pengcheng Bu (ID) http://orcid.org/0000-0002-3208-0354
Xiling Shen (ID) http://orcid.org/0000-0002-4978-3531

## Ethics
Animal experimentation: Mouse maintenance and procedures were approved by Duke University DLAR and followed the protocol (A286-15-10).

## Decision letter and Author response
Decision letter https://doi.org/10.7554/eLife.39479.032
Author response https://doi.org/10.7554/eLife.39479.033

## Additional files
### Supplementary files
• Transparent reporting form
DOI: https://doi.org/10.7554/eLife.39479.028

### Data availability
The RNA-seq data have been included as Figure 8-source data 1 and 2. They have also been deposited to GEO under the accession number GSE123628.

The following dataset was generated:

| Author(s) | Year | Dataset title | Dataset URL | Database and Identifier |
|---|---|---|---|---|
| L Wang, E Wang, Y Wang, R Mines, K Xiang, Z Sun, G Zhou, K Chen, S Chao, G Ye, H Yan, H Shan, J Everitt, P Bu, X Shen, N Rakhilin | 2018 | RNA-seq of Splenic CD4+ T cells and colon epithelial cells from miR-34a-/- and wildtype mice | https://www.ncbi.nlm.nih.gov/geo/query/acc.cgi?acc=GSE123628 | Gene Expression Omnibus (GEO), GSE123628 |

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
