## [Decision Letter]

[Editors’ note: a previous version of this study was rejected after peer review, but the authors submitted for reconsideration. The first decision letter after peer review is shown below.]

Thank you for choosing to send your work, "miR-34a is a microRNA safeguard for the inflammatory colon stem cell niche", for consideration at *eLife*. Your initial submission has been reviewed by three peer reviewers, one of whom is a member of our Board of Reviewing Editors, and the evaluation has been overseen by a Senior Editor. Although the work is of interest, we regret to inform you that the findings at this stage are too preliminary for further consideration at *eLife*.

Specifically, each of the three reviewers raised a number of major points. Reviewers 1 and 3 raise the need to provide histological and pathology analyses to better characterize the relation to tumor susceptibility. All reviewers commented that you haven't adequately shown that mi34 null mice are enriched for stem cells. This necessitates either FACS quantification of purified ISCs. Reviewer 1 raises a discrepancy in Figure 2 data that needs resolving. Reviewers 1 and 3 raised the need to know the relative expression levels of miR-34a and how they change. Reviewers 2 and 3 raise issues relating to your links to inflammation. Reviewer 3 indicates the need for CRISPR/CAS to complement the overexpression data, and this seems feasible in organoids and important to illuminate the specificity of your phenotype. Finally, reviewer 3 raises an important concern related to better controls needed to establish Tamoxifen efficiency. This reviewer also raises a key issue relating to conservation of the 3'UTR sites, and conversely, reviewer 1 raises an issue relating to the specificity to miR34a versus other family members. Addressing these issues would take considerable work, significantly more than the 2 months expected for revised manuscripts. However, the reviewers think the work is interesting, in principle, so a revised manuscript addressing these issues would be considered. Such a revised manuscript would be technically a "new" submission that undergoes the usual 2-step review, but the first step would be conducted by the reviewing editor and (usually) reviewers.

Reviewer #1:

Wang et al. reported that miR-34a functions in the colon to govern homeostasis and antagonize transformation upon bacteria infection. This occurs via both cell autonomous and non-autonomous mechanisms. Overall, if substantiated, the findings could be of broad appeal to the *eLife* readership. However, several important issues need to be addressed.

Figure 1: miR-34a-/- renders infected mice susceptible to colon tumorigenesis. The authors need to substantiate their conclusions with pathological and histological analysis to show the tumor stage and progression.

Figure 1: The increase of LGR5 and ASCL2 shown by Western: is this based on purified ISCs, or is it reflecting an increase from total colon tissue (reflective of increased cell number expressing LGR5 and ASCL2)?

Figure 2: It appears that miR-34a-/- alone does not significantly increase IL17 (Figure 2A-B). However from immunostaining (Figure 2C-D), miR-34a-/- uninfected colon clearly has increased IL17+ cells, whereas WT uninfected colon seems non-phenotypic. This discrepancy needs to be explained experimentally.

Figure 4: It appears that there is basal pSTAT3 activity in normal uninfected colon epithelium, and Stattic suppresses organoid formation/survival independent of IL-17. Is it true that Stattic treatment alone will give rise to a phenotype similar to IL-17+ Stattic? If so, it is difficult for a reviewer to accept a conclusion that pSTAT3 is the mediator of proliferation upon IL-17 stimulation.

Figure 8: Is miR-34a expression specific to the stem cells? What about the other miR-34 family members that might function analogously? What are the global gene expression changes in either epithelial or immune cells upon miR-34a acute deletion? What are miR-34a's direct targets besides the cherry-picked ones reported here? The authors need to address these issues.

Reviewer #2:

In their article "*mir-34a* is a microRNA safeguard for inflammatory colon stem cell niche" Shen and colleagues examine the role of mir-34a in limiting inflammatory pathology and tumorigenesis in following Citrobacter infection. In doing so they uncover a number of mir-34a targets in both immune and epithelial cells that synergize to promote pathology. I believe this is an intriguing study and is clinical relevant for a number of intestinal inflammatory conditions. However I have a few concerns regarding controls and interpretations of experiments that should be addressed prior to consideration for publication in *eLife*.

• The authors claim that the crypts of infected mir34ko animals were enriched for stem cells based on increased Lgr5 and Ascl2 expression. Without lineage tracing stem cell it is difficult to make this claim as differentiated cells could merely be up-regulating the expression of these stem cell markers. The title of this manuscript may also need to be modified to reflect this.

• The authors claim that the 3' UTR of IL23R has a mir34a-binding site, in the absence of which there is more IL-17A production from CD4 T cells that contributes to pathology. What are the absolute numbers of these pathological Th17 cells? It is difficult to claim there are more Th17 cells based on frequencies on a FACS plot. Were they acquired from the gut or draining lymph node? Other cell types also express IL23R, for example ILCs. What is their level of IL-17A expression? Further what is the level of IL23R expression in the mir34aKo animals?

• The authors test a role for stat3 in their model and claim that IL-17A activates stat3. Since IL17R's are known to signal through NF-κB, are these receptors also activating stat3 or is this a secondary effect of a downstream cytokine (for example IL-6 or IL-24) being produced?

• What is the level of epithelial IL17RD expression in mir34aKO animals?

• Does treatment of infected mir-34a KO with anti-IL17A abrogate pathology and tumorigenesis?

Reviewer #3:

In this manuscript, Wang et al. propose a novel function for miR-34a, a member of a family of p53-regulated miRNAs, in controlling the inflammatory response of the colon to infection by *Citrobacterium rodentium*.

They present a model in which miR-34a directly represses IL23R and CCL22 in CD4^+^ cells and IL17RD in the intestinal cells. Repression of these targets by miR-34a is necessary to prevent excessive Th17 differentiation and migration, and to blunt the proliferative response of the intestinal stem cells to IL17. By doing so, miR-34a acts as a barrier against inflammation-induced intestinal tumorigenesis. These findings are of substantial relevance and broad interest to the readership of *eLife*.

The manuscript is clearly written, the experiments are well described, and their results are largely consistent with the model proposed by the authors. Major strengths include the use of genetically engineered mice harboring targeted deletion of miR-34a and the use of intestinal organoids derived from these mice to explore the mechanisms through which miR-34a affects intestinal tumorigenesis and inflammation.

Despite these major strengths, the manuscript has a few weaknesses that in my opinion need to be addressed. More specifically:

1) One of the central claims of the authors is that loss of miR-34a leads to high incidence of colon cancers in *Citrobacterium*-infected mice. This claim is not adequately supported by the evidence presented in the manuscript. The only analysis provided in addition to the statement that 11 out of 20 mice developed tumors, are a single H&E staining (Figure 1A) and a single macroscopic image of a dissected colon (Figure 1B). Given the importance of this claim much more details should be provided. In particular, the authors should include a detailed histopathologic characterization of the lesions by an experience animal pathologist. This should be corroborated by low and high magnification images, immunostaining for β catenin, and histologic grading. Additional key questions that should be addressed include: Are the tumors invasive/metastatic? Are they adenocarcinomas or adenomas? Are they APC mutant? The pro-tumorigenic effect of miR-34 loss in this context is a major novelty of the manuscript, and therefore I believe these are important details that need to be included to support and better define the claim.

2) A more detailed characterization of the dynamic of inflammation in response to *Citrobacterium* infection in miR-34a wt and mutant mice should be provided. The authors only show the consequences at 11 days and at 6 months (Figure 1A), but it would be important to know what the intestine looks at intermediate time points. Does a state of chronic inflammation and intestinal stem cell hyper-proliferation develop in the miR-34a-null mice that would explain the increased tumor incidence at later time points?

3) Another essential aspect that needs to be addressed concerns the expression dynamic of miR-34a in the intestinal epithelium and in CD4^+^ cells undergoing Th17 differentiation. Knowing the relative expression levels of miR-34a, and how they change in response to *Citrobacterium* infection is essential to assess the plausibility of the proposed mechanism.

4) The authors propose that miR-34a directly targets IL17RD, IL23R, and CCL22 by binding to their respective 3'UTRs. In support of this hypothesis they present data that luciferase reporters containing the wt or mutant 3'UTR are differentially sensitive to miR-34a overexpression. While this is a reasonable idea, it is important to recognize that in these experiments the ectopically expressed miRNA achieves levels that are often hundred folds greater than their physiologic levels. A more conclusive and elegant experiment would be to use CRISPR-Cas9 to introduce a point mutation in the predicted binding site and show that this leads to upregulation of the gene product in mIR-34a wt, but not in miR-34a-null cells. This experiment should be relatively straightforward to perform to validate the IL17RD site in intestinal organoids derived from miR-34a wild type and mutant mice.

5) An additional reason I am not fully convinced that miR-34a targets these genes directly via the proposed sites is that none of the three 3'UTR sites is conserved in the human genome. The authors unfortunately fail to discuss this important point in the manuscript, a point that is even more relevant since they suggest that miR-34a plays a similar mechanism in suppressing colon cancer formation in humans (Figure 7).

6) Finally, the experiments described in Figure 6 using conditional deletion of miR-34a in the intestine with Lgr5-Cre-ER are elegant but difficult to interpret because the authors do not show how efficiently tamoxifen delivery lead to bi-allelic miR-34a deletion in the intestinal epithelium. Poor Cre-mediated recombination could be a simple explanation for the reduced tumor incidence. This should be addressed if the authors want to include this experiment in the manuscript.

Another useful control would be to include a cohort of mice generated by transplanting the bone marrow from miR-34a-/- mice into tamoxifen-treated Lgr5-Cre-ER;miR-34a fl/fl recipients. If the model proposed by the authors is correct, these animals should display a tumor incidence in response to *Citrobacterium* infection analogous to what was observed in miR-34a-/- animals. I understand, however, that performing these experiments would be a major undertaking in terms of time and resources and I would not insist on having this latter experiment included in the revised paper.

[Editors’ note: what now follows is the decision letter after the authors submitted for further consideration.]

Thank you for submitting your article "*miR-34a* is a microRNA safeguard for inflammatory colon oncogenesis" for consideration by *eLife*. Your article has been reviewed by 3 peer reviewers, including Elaine Fuchs as the Reviewing Editor and Reviewer #1, and the evaluation has been overseen Kevin Struhl as the Senior Editor. The following individual involved in review of your submission has agreed to reveal their identity: Andrea Ventura (Reviewer #3).

The reviewers have discussed the reviews with one another and the Reviewing Editor has drafted this decision to help you prepare a revised submission.

The reviewers overall found your manuscript to be substantially improved and in principle of interest to *eLife* readership. However, there are still several concerns, which in our view are significantly substantial that we feel will need addressing experimentally. Normally, we do not pursue with the process if we feel that it could take the authors more than 2 months to complete. However, given the enthusiasm for your paper, we are willing to leave this door open for you, and encourage you to address the issues below, add the experiments, revise the text and figures and resubmit. If you address these final issues, I can ensure a rapid turnaround, but please do not take our enthusiasm for your work as an acceptance of your manuscript. The issues below need to be dealt with in a satisfactory way.

The main lingering issue centers on the quality of some of your data and the lack of appropriate controls, which call into question your interpretation. Specifically, you claim that the crypts of infected mir34ko animals are enriched for stem cells based on increased Lgr5 and Ascl2 expression. Without either lineage tracing the stem cells or some kind of functional assay, this claim is not substantiated. Differentiated cells could merely be up-regulating the expression of these stem cell markers. The title of your paper is not justified without these data.

An additional concern raised by reviewer 2 is that you claim that the 3' UTR of IL23R and IL16R has a mir34a-binding site, in the absence of which there is more IL17 production from CD4 T cells that contributes to pathology. What are the absolute numbers of these pathological Th17 cells? It is difficult to claim there are more Th17 cells based on frequencies on a FACS plot. Were they acquired from the gut or draining lymph node? Other cell types also express IL23R, for example ILCs. What is their level of IL-17A expression? Further what is the level of IL23R expression in the mir34aKo animals?

Of additional concern, your team has tested for a role for stat3 and claim that IL17A activates stat3. Since IL17R's are known to signal through NF-κB, are these receptors also activating stat3 or is this a secondary effect of a downstream cytokine (for example IL-19/20/ IL-24) being produced? Again, it is difficult to make this link without the underlying mechanisms or at least a possible discussion of one. Related to this, what is the level of epithelial IL17RD expression in mir34aKO animals? Does treatment of infected mir-34a KO with anti-IL17A abrogate pathology and tumorigenesis?

In your Lgr5-GFP-CreER; mIR34a fl/fl mouse model, how are the mRNA or protein levels of proposed mIR34a downstream targets affected? CCL22, IL17 etc. (Figure 6A). Figure 6D is lacking the ko>ko and WT>WT BM chimera control. It is difficult to interpret the use of WT controls, which have not undergone irradiation (a treatment known to affect the intestinal compartment).

Finally, reviewer 3 points out that the volcano plots shown in the supplementary figure should be complemented by including cumulative distribution (cdf) plots with logFC on the x axis comparing predicted miR34a targets to the rest of the transcriptome. These plots should show preferential derepression of the miR-34a targets (i.e. a shift to the right of the cdf plot compared to background). These are standard plots in the miRNA field and the reason for asking to include them has to do with the initial concerns raised regarding the absolute levels of miR-34a in these tissues. Demonstrating a signal attributable to miR-34a loss in the cdf plots would be conclusive evidence that the levels of miR-34a in these tissues are high enough to cause substantial target repression.

[Editors' note: further revisions were requested prior to acceptance, as described below.]

Thank you for resubmitting your work entitled "*miR-34a* is a microRNA safeguard for *Citrobacter*-induced inflammatory colon oncogenesis" for further consideration at *eLife*. Your revised article has been reviewed by three peer reviewers, one of whom is a member of our Board of Reviewing Editors, and the evaluation has been overseen by Kevin Struhl as the Senior Editor.

The manuscript has been improved but there is still one issue that can be addressed textually before acceptance, as outlined below:

Reviewer 2 makes a valid point, namely that in Figure 2—figure supplement 1 you claim to evaluate ILC3 based on CD117+ CD45+ cells. As this is not a specific straining strategy for ILC3, you should revise your claim that ILC3s are not affected in your system. Once this is addressed textually, there should be no further need for adjustments.

The set of reviews are listed below:

Reviewer #1:

The reviewers have addressed our concerns satisfactorily, and in our view, the manuscript is now suitable for publication in *eLife*.

Reviewer #2:

The revised manuscript is significantly improved. The newly included experiments have validated many of the author's initial claims and changes to the text have corrected over statements. I believe the study is well suited for publication in e*Life*.

I have one comment that can be addressed textually.

In Figure 2—figure supplement 1 the authors claim to evaluate ILC3 based on CD117+ CD45+ cells. This is in no way a specific straining strategy for ILC3 and the authors should revise their claims of ILC3s not being effected in their system based on this rather rudimentary analysis.

Reviewer #3:

I am satisfied with the changes made by the authors, who have addressed my remaining concerns.

---

## [Author Response]

[Editors’ note: the author responses to the first round of peer review follow.]

Reviewer #1:Wang et al. reported that miR-34a functions in the colon to govern homeostasis and antagonize transformation upon bacteria infection. This occurs via both cell autonomous and non-autonomous mechanisms. Overall, if substantiated, the findings could be of broad appeal to the eLife readership. However, several important issues need to be addressed.Figure 1: miR-34a-/- renders infected mice susceptible to colon tumorigenesis. The authors need to substantiate their conclusions with pathological and histological analysis to show the tumor stage and progression.

We agree with reviewer that pathological and histological analysis is needed to substantiate our conclusion. Dr. Jeffrey Everitt, a board-certified veterinary pathologist with extensive murine tumor experience, reviewed whole slide scanning of H&E stained colon sections from day 11, month 2, 4, and 6 post infection. His review confirmed that colon tumorigenesis and progression in the infected miR-34a -/- KO mice was histologically similar to those noted in other murine colon carcinogenesis models [1] (Figure 1A and Figure 1—figure supplement 1A).

In detail, histopathological changes of pre-neoplasia and neoplasia were limited to the miR-34a /- genotype and were first noted at the four-month time point. Microscopic sections from wild type control mice were free of dysplastic and neoplastic changes at four-month (0/2) and six-month time points (0/20) following infection. In miR34a-/- mice no dysplasia or early neoplasia was present at a two-month time point (0/4), whereas at four months half the animals (2/4) had dysplastic change microscopically. At the six-month time point (11/20) miR-34a-/- mice had microscopic changes ranging from dysplasia (2/20), to adenoma (7/20), to adenocarcinoma (2/20). The tumors were considered well-differentiated. Interestingly, the animal with a colonic adenocarcinoma in the distal colon also had a squamous cell carcinoma of the rectum.

We also checked the livers and lungs of the infected miR-34a-/- mice, and found no metastases (Figure 1—figure supplement 1B, C).

Figure 1: The increase of LGR5 and ASCL2 shown by Western: is this based on purified ISCs, or is it reflecting an increase from total colon tissue (reflective of increased cell number expressing LGR5 and ASCL2)?

Western blots were performed on the total colon tissue.

Figure 2: It appears that miR-34a-/- alone does not significantly increase IL17 (Figure 2A-B). However from immunostaining (Figure 2C-D), miR-34a-/- uninfected colon clearly has increased IL17+ cells, whereas WT uninfected colon seems non-phenotypic. This discrepancy needs to be explained experimentally.

In Figure 2C-D, we only examined infected colon from WT and miR-34a-/- mice (marked WT infected and miR-34a-/- infected). Figure 2C showed CD4 staining and Figure 2D showed IL-17 staining of infected colon tissue, not uninfected colon.

Figure 4: It appears that there is basal pSTAT3 activity in normal uninfected colon epithelium, and Stattic suppresses organoid formation/survival independent of IL-17. Is it true that Stattic treatment alone will give rise to a phenotype similar to IL-17+ Stattic? If so, it is difficult for a reviewer to accept a conclusion that pSTAT3 is the mediator of proliferation upon IL-17 stimulation.

We treated the colon organoids with 20μM Stattic along with different amounts of IL-17A (2ng, 4ng and 8ng). Stattic alone significantly suppressed colon organoids growth and pSTAT3 activity. 2ng IL-17A mitigated the effect of Stattic, 4ng IL-17A rescued colon organoids growth and pSTAT3 activity, and 8ng IL-17A further enhanced colon organoids growth and pSTAT3 activity (Figure 4—figure supplement 7).

The original manuscript only included the treatment condition of 20μM Stattic and 2ng IL-17A (Figure 4E-G). In the revised manuscript, we have clarified this condition in the Figure 4 legend and included all conditions in Figure 4—figure supplement 7 to provide readers with the complete picture.

Figure 8: Is miR-34a expression specific to the stem cells? What about the other miR-34 family members that might function analogously? What are the global gene expression changes in either epithelial or immune cells upon miR-34a acute deletion? What are miR-34a's direct targets besides the cherry-picked ones reported here? The authors need to address these issues.

We isolated Lgr5-GFP+ and Lgr5-GFP- cells by FACS and measured their respective miR-34a levels by RT-qPCR. miR-34a expression is three-fold higher in Lgr5-GFP+ stem cells than in Lgr5-GFP- cells (Figure 5—figure supplement 9A).

We also measured the levels of miR-34b and miR-34c, the other two members in the miR-34a family, in colon epithelial cells by RT-qPCR. miR-34b and miR-34c expression levels were barely detectable, about one hundred fold lower than miR-34a (Figure 5—figure supplement 9B).

We performed RNA-seq on splenic CD4^+^ T cells and colon epithelial cells isolated from miR34a-/- and wildtype mice, which revealed global gene expression changes (Figure 8—figure supplement 13 and Figure 8—source data 1 and 2). It is worth noting that RNA-seq tends to pick up more abundantly expressed genes depending on the sequencing depth. The known direct miR34a target genes such as Notch1, Snail2, BCL2, and c-Met were upregulated in miR-34a-/- cells. They could have certainly contributed to tumorigenesis as previously reported. However, because the uninfected miR-34a-/- mice did not form spontaneous tumors, these genes alone were not sufficient to cause tumorigenesis without the immune system. We have added the following sentence to the Discussion section of the manuscript:

“miR-34a is a known tumor suppressor that targets cell proliferation and apoptosis genes. In fact, RNA-seq performed on splenic CD4^+^ T cells and colon epithelial cells isolated from miR-34a-/- and wildtype mice revealed various changes in gene expression (Figure 8—figure supplement 13), including well-known miR-34a target genes such as Notch1, Snail, BCL2, and c-Met (Figure 8—source data 1 and 2). However, miR-34a deficiency alone does not lead to tumorigenesis, suggesting that mere upregulation of these genes by miR-34a loss is not sufficient to cause cancer.”

Interestingly, IL-6R was also shown upregulated in CD4^+^ T cells. The IL-6/IL-6R axis plays an important role in Th17 cell differentiation. IL-6R has previously been reported to be a miR-34a target in epithelial cells [2] but whether the miR-34a-IL-6R axis plays any role in immune cells has not been reported. We validated that miR-34a directly targets IL6R in CD4^+^ T cells, which suggests that miR-34a targets both IL-6R and IL-23R to modulate Th17 cell differentiation (Figure 3). This finding that miR-34a targets both steps (IL-6R and IL-23R) in Th17 differentiation strengthens our conclusion that miR-34a is an important regulator for the inflammatory niche, and has been added to the revised manuscript.

Reviewer #2:In their article "mir-34a is a microRNA safeguard for inflammatory colon stem cell niche" Shen and colleagues examine the role of mir-34a in limiting inflammatory pathology and tumorigenesis in following Citrobacter infection. In doing so they uncover a number of mir-34a targets in both immune and epithelial cells that synergize to promote pathology. I believe this is an intriguing study and is clinical relevant for a number of intestinal inflammatory conditions. However I have a few concerns regarding controls and interpretations of experiments that should be addressed prior to consideration for publication in eLife.• The authors claim that the crypts of infected mir34ko animals were enriched for stem cells based on increased Lgr5 and Ascl2 expression. Without lineage tracing stem cell it is difficult to make this claim as differentiated cells could merely be up-regulating the expression of these stem cell markers. The title of this manuscript may also need to be modified to reflect this.

We thank reviewer for the rigor. We have modified the title to: *miR-34a* is a microRNA safeguard for inflammatory colon oncogenesis.

• The authors claim that the 3' UTR of IL23R has a mir34a-binding site, in the absence of which there is more IL-17A production from CD4 T cells that contributes to pathology. What are the absolute numbers of these pathological Th17 cells? It is difficult to claim there are more Th17 cells based on frequencies on a FACS plot. Were they acquired from the gut or draining lymph node? Other cell types also express IL23R, for example ILCs. What is their level of IL-17A expression? Further what is the level of IL23R expression in the mir34aKo animals?

After CD4 T cell isolation, we counted 10,000 CD4^+^ T cells for each FACS analysis. According to the percentage of the interested cell population of FACS analysis, 710 out of 10,000 were IL17+ cells in the infected wildtype colon tissue while 2410 out of 10,000 were IL-17+ cells in the infected miR-34a-/- colon tissue, as indicated by the percentages on the FACS plots (Figure 2B). We have clarified the total CD4^+^ T cell numbers in the figure legend and the methods under the section title “CD4^+^ T cell isolation and Th17 cell differentiation”.

The CD4^+^ cells were acquired from the gut. We have clarified this in the main text and figure legend.

The protocol we found for isolating ILCs requires lineage markers (CD3, CD5, CD19, Ly6 C/G), surface staining makers (IL7α, CD45, CCR6) and transcription factors (RORγ, GATA3). The sorting results was limited by our flow cytometry ability. We attempted twice without convincing results. Furthermore, to characterize IL23R and IL-17A levels in ILCs at different post-infection points will require repeating of the *Citrobacter* infection experiments and will take more than 8 months (6 months for the tumors to form). Therefore, we were not able to complete this experiment. We have added this caveat that ILCs also express IL23R and may play a role in the observed phenotype to our Discussion section as follows:

“Furthermore, besides Th17 cells, innate lymphoid cells (ILCs) may also influence the stem cell niche as IL-23R+ ILCs have been reported to increase during inflammatory bowel disease and are capable of inducing colitis [Baker et al., 2015; Geremia et al., 2011].”

miR-34a knockout increased IL-23R expression in CD4^+^ T cells from uninfected animals (Figure 3—figure supplement 4A).

• The authors test a role for stat3 in their model and claim that IL-17A activates stat3. Since IL17R's are known to signal through NF-κB, are these receptors also activating stat3 or is this a secondary effect of a downstream cytokine (for example IL-6 or IL-24) being produced?

We treated the colon organoids with 20μM Stattic along with different amount of IL-17A (2ng, 4ng and 8ng). Stattic alone significantly suppressed colon organoids growth and pSTAT3 activity. 2ng IL-17A could abrogate Stattic effect, while 4ng IL-17A could rescue colon organoids growth and pSTAT3 activity and 8ng IL-17A further enhanced colon organoids growth and pSTAT3 activity (Figure 4—figure supplement 7). Therefore, we think pSTAT3 activity is required to maintain colon organoids growth and IL-17 stimulation promotes organoids survival at least partially through pSTAT3 activity. We included the results in the revised manuscript.

As the reviewer mentioned, IL-17A/IL-17RA could enhance NF-κB signaling and upregulate IL6 expression which induces stat3 activation ([3]). In our study, both IL-17RA and IL-17RD are important for colon organoids growth. It is not clear whether IL-17/IL-17RA/IL-17RD directly activate stat3. However, we think it is likely NF-κB plays an important role in stat3 activation and colon organoids growth. We have added the following discussion to our revised manuscript: “IL-17A/IL-17RA could enhance NF-_κ_B signaling and upregulate IL-6 expression, which induces stat3 activation [Wang et al., 2014]. Therefore, NF-_κ_B may play an important role in the epithelial response to IL-17 stimulation.”

• What is the level of epithelial IL17RD expression in mir34aKO animals?

Loss of miR-34a increased IL-17RD expression according to western blot (Figure 3—figure supplement 4B).

• Does treatment of infected mir-34a KO with anti-IL17A abrogate pathology and tumorigenesis?

We had anticipated this important question and were in the middle of this experiment when we submitted our first version. After infection, we treated the mice with anti-IL17 neutralizing antibody in the first 2 months, stopped treatment in the next 4 months, and sacrificed the mice at the end of month 6 post infection. The anti-IL17 treatment suppressed tumorigenesis (Figure 7JL), consistent with its effect on suppressing stem cell proliferation (Figure 7G-I). This result has been added to the revised manuscript.

Reviewer #3:[…] The manuscript is clearly written, the experiments are well described, and their results are largely consistent with the model proposed by the authors. Major strengths include the use of genetically engineered mice harboring targeted deletion of miR-34a and the use of intestinal organoids derived from these mice to explore the mechanisms through which miR-34a affects intestinal tumorigenesis and inflammation.Despite these major strengths, the manuscript has a few weaknesses that in my opinion need to be addressed. More specifically:1) One of the central claims of the authors is that loss of miR-34a leads to high incidence of colon cancers in Citrobacterium-infected mice. This claim is not adequately supported by the evidence presented in the manuscript. The only analysis provided in addition to the statement that 11 out of 20 mice developed tumors, are a single H&E staining (Figure 1A) and a single macroscopic image of a dissected colon (Figure 1B). Given the importance of this claim much more details should be provided. In particular, the authors should include a detailed histopathologic characterization of the lesions by an experience animal pathologist. This should be corroborated by low and high magnification images, immunostaining for β catenin, and histologic grading. Additional key questions that should be addressed include: Are the tumors invasive/metastatic? Are they adenocarcinomas or adenomas? Are they APC mutant? The pro-tumorigenic effect of miR-34 loss in this context is a major novelty of the manuscript, and therefore I believe these are important details that need to be included to support and better define the claim.

We completely agree with reviewer that a detailed histopathologic characterization is needed. In fact, reviewer 1 has raised similar concerns (see his question 1). Dr. Jeffrey Everitt, a board certified veterinary pathologist with extensive murine tumor experience, reviewed whole slide scanning of H&E stained colon sections from day 11, month 2, 4, and 6 post infection. His review confirmed that colon tumorigenesis and progression in the infected miR-34a -/- KO mice was histologically similar to those noted in other murine colon carcinogenesis models [1] (Figure 1A and Figure 1—figure supplement 1A). Immunostaining for β-catenin is consistent with neoplastic progression (Figure 1A).

In detail, histopathological changes of pre-neoplasia and neoplasia were limited to the miR 34a -/- genotype and were first noted at the four-month time point. Microscopic sections from wild type control mice were free of dysplastic and neoplastic changes at four-month (0/2) and six-month time points (0/20) following infection. In miR34a -/- mice no dysplasia or early neoplasia was present at a two-month time point (0/4), whereas at four months half the animals (2/4) had dysplastic change microscopically. At the six month time point (11/20) miR-34a-/- mice had microscopic changes ranging from dysplasia (2/20), to adenoma (7/20), to adenocarcinoma (2/20). The tumors were considered well-differentiated. Interestingly, the animal with a colonic adenocarcinoma in the distal colon also had a squamous cell carcinoma of the rectum.

The β-catenin IHC shows that dysplastic and neoplastic changes were characterized by strong intracytoplasmic staining and occasional cells with nuclear staining (Figure 1A).

We checked liver and lung of the infected miR-34a-/- mice with colon tumors. No metastasis was found in liver and lung, the most frequency metastatic organs for colon tumors (Figure 1—figure supplement 1B, C).

We checked APC mutation on saved tumors. We designed primers to amplify three most APC mutation regions in mouse colon cancer [4-7]. DNA sequencing showed the two regions were not mutated in the miR-34a-/- tumors (Figure 1—figure supplement 2).

2) A more detailed characterization of the dynamic of inflammation in response to Citrobacterium infection in miR-34a wt and mutant mice should be provided. The authors only show the consequences at 11 days and at 6 months (Figure 1A), but it would be important to know what the intestine looks at intermediate time points. Does a state of chronic inflammation and intestinal stem cell hyper-proliferation develop in the miR-34a-null mice that would explain the increased tumor incidence at later time points?

We analyzed the colon cancer progression on 11 days, 2 month, 4 months and 6 months after *Citrobacterium* infection. Images from all four time points have been included in the revised manuscript (Figure 1A and Figure 1—figure supplement 1A).

We agree with the reviewer that a state of chronic inflammation and stem cell hyper-proliferation explains the increased tumor incidence. In miR34a-/- mice, at four months half the animals (2/4) had inflammation-induced dysplastic change and at six months, half the animals (11/20) formed tumors. According to the pathologist: “the earliest dysplastic changes are noted in deep reaches of crypts that are within inflamed ulcerated colonic mucosa in several of the sections where the diffuse inflammation of the *Citrobacter* has subsided and focal long-standing inflammation has set up due to ulceration of the surface. Further links the role of inflammation and carcinogenesis.”

3) Another essential aspect that needs to be addressed concerns the expression dynamic of miR-34a in the intestinal epithelium and in CD4^+^ cells undergoing Th17 differentiation. Knowing the relative expression levels of miR-34a, and how they change in response to Citrobacterium infection is essential to assess the plausibility of the proposed mechanism.

We infected mice with *Citrobacter* and collected CD4^+^ T cells and colon epithelial cells on day 7, day 14 and day 21 post-infection, followed by RT-qPCR measurements. miR-34a expression levels were upregulated in both CD4^+^ T cells and colon epithelial cells after infection (Figure 5—figure supplement 9C, D).

4) The authors propose that miR-34a directly targets IL17RD, IL23R, and CCL22 by binding to their respective 3'UTRs. In support of this hypothesis they present data that luciferase reporters containing the wt or mutant 3'UTR are differentially sensitive to miR-34a overexpression. While this is a reasonable idea, it is important to recognize that in these experiments the ectopically expressed miRNA achieves levels that are often hundred folds greater than their physiologic levels. A more conclusive and elegant experiment would be to use CRISPR-Cas9 to introduce a point mutation in the predicted binding site and show that this leads to upregulation of the gene product in mIR-34a wt, but not in miR-34a-null cells. This experiment should be relatively straightforward to perform to validate the IL17RD site in intestinal organoids derived from miR-34a wild type and mutant mice.

We cloned the gRNA:CAGAATGATGGCGGTGGCAG-TGG (sequence complementary to miR-34a binding site highlighted by red) into the vector pLentiCRISPR v2, and transfected the CRISPR/CAS9 vector into mouse colon organoids). DNA sequencing of single colonies confirmed successful deletion of miR-34a binding in the mouse IL-17RD 3`UTR. Western blot showed that miR-34a binding site deletion increased IL-17RD expression in wildtype mouse organoids but not in miR-34a-/- organoids (Figure 5—figure supplement 8).

Unfortunately we could not find any PAM motif around the miR-34a binding site in the CCL22 3`UTR (cagtgactgccacagtttgttggtat) or in the IL23R 3`UTR (ataagtacactgccatgactccaggatg), so we were not able to mutate these two binding sites specifically.

5) An additional reason I am not fully convinced that miR-34a targets these genes directly via the proposed sites is that none of the three 3'UTR sites is conserved in the human genome. The authors unfortunately fail to discuss this important point in the manuscript, a point that is even more relevant since they suggest that miR-34a plays a similar mechanism in suppressing colon cancer formation in humans (Figure 7).

We thank the reviewer for this important question. The reviewer is correct that the 3`UTR sites are not conserved between mouse and human. We validated miR-34a binding sites in human IL-17RD, CCL22 and IL-6R 3UTR, and the binding site sequences are different between human and mice.

We ectopically expressed miR-34a in SW480 cells, a human colon cancer cell line and Jurkat cells, a human lymphocyte cell line, then measured IL-17RD and CCL22 expression in SW480 cells and IL-23R in Jurkat cells. The results showed that miR-34a suppressed IL-17RD and CCL22 expression, but not IL-23R (Figure 7—figure supplement 12). We added this into the Discussion section.

Interestingly, RNA-seq suggested by reviewer 1 showed that IL-6R was upregulated in CD4^+^ T cells. The IL-6/IL-6R axis plays important roles in Th17 cell differentiation, similar to IL-23. We validated that miR-34a directly targets IL6R in mouse CD4^+^ T cells, which suggests that miR34a targets both IL-6R and IL-23R to modulate mouse Th17 cell differentiation (Figure 3). We hypothesized that in human, miR-34a may modulate Th17 differentiation by targeting IL-6R alone. Indeed, ectopic miR-34a expression upregulated IL-6R expression in Jurkat cells (Figure 7—figure supplement 12).

We then checked the miR-34a binding sites in the 3`UTRs of the human genes using miRanda and RNA22. Luciferase reporter assay showed that miR-34a directly binds to its putative binding sites in human IL-6R, CCL22, and IL-17RD 3`UTRs, but not in the human IL-23R 3`UTR (Figure 7—figure supplement 12).

Collectively, in mice miR-34a targets IL-6R and IL-23R simultaneously to regulate Th17 differentiation, and target CCL22 and IL-17RD in epithelial cells (Figure 8). In human, miR-34a only targets IL-6R but not IL-23R in T cells, and still targets CCL22 and IL-17RD in epithelial cells. We thank the reviewers’ thoughtful questions, which has helped us reveal the role of IL-6R in this process. This provides an insight that, although miR-34a targeting of IL-23R is not conserved between mice and human, its function of regulating Th17 differentiation is still conserved via targeting IL-6R. These experiments have greatly strengthened the relevance of our study to the human disease.

6) Finally, the experiments described in Figure 6 using conditional deletion of miR-34a in the intestine with Lgr5-Cre-ER are elegant but difficult to interpret because the authors do not show how efficiently tamoxifen delivery lead to bi-allelic miR-34a deletion in the intestinal epithelium. Poor Cre-mediated recombination could be a simple explanation for the reduced tumor incidence. This should be addressed if the authors want to include this experiment in the manuscript.Another useful control would be to include a cohort of mice generated by transplanting the bone marrow from miR-34a-/- mice into tamoxifen-treated Lgr5-Cre-ER;miR-34a fl/fl recipients. If the model proposed by the authors is correct, these animals should display a tumor incidence in response to Citrobacterium infection analogous to what was observed in miR-34a-/- animals. I understand, however, that performing these experiments would be a major undertaking in terms of time and resources and I would not insist on having this latter experiment included in the revised paper.

We validated miR-34a deletion after tamoxifen treatment by genotyping, which has been included in the revised manuscript (Figure 6—figure supplement 10). We agree with the reviewer that it would be an elegant experiment to transplant bone marrow from miR-34a-/- mice into tamoxifen-treated Lgr5-Cre-ER / miR-34a fl/fl mice. However, because we do not have enough Lgr5-Cre-ER / miR-34a fl/fl mice on hand, we would have to breed them first, and the *Citrobacter* infection experiment will take more than 6 months to conclude, so this experiment will likely take at least a year to complete. We thank the reviewer for the understanding.

Reference:

1) M.K. Washington, A.E. Powell, R. Sullivan, J.P. Sundberg, N. Wright, R.J. Coffey, W.F. Dove, Pathology of rodent models of intestinal cancer: progress report and recommendations, Gastroenterology, 144 (2013) 705-717.

2) M. Rokavec, M.G. Oner, H. Li, R. Jackstadt, L. Jiang, D. Lodygin, M. Kaller, D. Horst, P.K. Ziegler, S. Schwitalla, J. Slotta-Huspenina, F.G. Bader, F.R. Greten, H. Hermeking, Corrigendum. IL6R/STAT3/miR-34a feedback loop promotes EMT-mediated colorectal cancer invasion and metastasis, J Clin Invest, 125 (2015) 1362.

3) K. Wang, M.K. Kim, G. Di Caro, J. Wong, S. Shalapour, J. Wan, W. Zhang, Z. Zhong, E. SanchezLopez, L.W. Wu, K. Taniguchi, Y. Feng, E. Fearon, S.I. Grivennikov, M. Karin, Interleukin-17 receptor a signaling in transformed enterocytes promotes early colorectal tumorigenesis, Immunity, 41 (2014) 1052-1063.

4) J. Ju, J. Hong, J.N. Zhou, Z. Pan, M. Bose, J. Liao, G.Y. Yang, Y.Y. Liu, Z. Hou, Y. Lin, J. Ma, W.J. Shih, A.M. Carothers, C.S. Yang, Inhibition of intestinal tumorigenesis in Apcmin/+ mice by (-)epigallocatechin-3-gallate, the major catechin in green tea, Cancer research, 65 (2005) 1062310631.

5) A.J. Macfarlane, C.A. Perry, M.F. McEntee, D.M. Lin, P.J. Stover, Shmt1 heterozygosity impairs folate-dependent thymidylate synthesis capacity and modifies risk of Apc(min)mediated intestinal cancer risk, Cancer research, 71 (2011) 2098-2107.

6) E.C. Robanus-Maandag, P.J. Koelink, C. Breukel, D.C. Salvatori, S.C. Jagmohan-Changur, C.A. Bosch, H.W. Verspaget, P. Devilee, R. Fodde, R. Smits, A new conditional Apc-mutant mouse model for colorectal cancer, Carcinogenesis, 31 (2010) 946-952.

7) A. Lorenz, M. Deutschmann, J. Ahlfeld, C. Prix, A. Koch, R. Smits, R. Fodde, H.A. Kretzschmar, U. Schuller, Severe alterations of cerebellar cortical development after constitutive activation of Wnt signaling in granule neuron precursors, Molecular and cellular biology, 31 (2011) 3326-3338.

[Editors' note: the author responses to the re-review follow.]

The reviewers overall found your manuscript to be substantially improved and in principle of interest to eLife readership. However, there are still several concerns, which in our view are significantly substantial that we feel will need addressing experimentally. Normally, we do not pursue with the process if we feel that it could take the authors more than 2 months to complete. However, given the enthusiasm for your paper, we are willing to leave this door open for you, and encourage you to address the issues below, add the experiments, revise the text and figures and resubmit. If you address these final issues, I can ensure a rapid turnaround, but please do not take our enthusiasm for your work as an acceptance of your manuscript. The issues below need to be dealt with in a satisfactory way.The main lingering issue centers on the quality of some of your data and the lack of appropriate controls, which call into question your interpretation. Specifically, you claim that the crypts of infected mir34ko animals are enriched for stem cells based on increased Lgr5 and Ascl2 expression. Without either lineage tracing the stem cells or some kind of functional assay, this claim is not substantiated. Differentiated cells could merely be up-regulating the expression of these stem cell markers. The title of your paper is not justified without these data.

We performed a functional assay by measuring organoid-forming efficiency (1, 2). Wildtype and miR-34a-/- mice were infected with *C. rodentium* for 2 months along with the uninfected control groups. The mouse colons were then collected for organoid culture. 1000 organoid cells from each condition were reseeded to measure the organoids-forming efficiency. Cells from *C. rodentium* infected miR-34a-/- mice had significantly higher organoid-forming efficiency and growth rates than the other groups (Figure 1F-G).

An additional concern raised by reviewer 2 is that you claim that the 3' UTR of IL23R and IL16R has a mir34a-binding site, in the absence of which there is more IL17 production from CD4 T cells that contributes to pathology. What are the absolute numbers of these pathological Th17 cells? It is difficult to claim there are more Th17 cells based on frequencies on a FACS plot. Were they acquired from the gut or draining lymph node? Other cell types also express IL23R, for example ILCs. What is their level of IL-17A expression? Further what is the level of IL23R expression in the mir34aKo animals?

We infected wildtype and miR-34a-/- mice with *C. rodentium* for 2 months and then measured the absolute numbers of Th17 (CD4^+^IL17+) cells in the mouse colons. According to the protocol established by Weigmann B, et. Al (3), the average Th17 cell numbers were 277, 313, 1163 and 3378 for uninfected wild-type, uninfected miR-34a-/-, infected wild-type and infected miR-34a-/- respectively (Figure 2B). The wild-type Th17 cell number is comparable with some previously published data (4), but fewer than other reports (5). The discrepancy (loss of cells) could be caused by the exact handling of cell isolation, Th17 staining, enzyme incubation time, and FACS analysis. Therefore the trend between the groups is probably more important.

The CD4^+^cells were acquired from the colon, not the draining lymph nodes. We have clarified this in the main text and figure legend.

Among the ILCs, ILC3 cells were reported to express IL17A 6). We infected wildtype and miR-34a-/- mice with *C. rodentium* for 2 months, then isolate ILC3 cells by FACS sorting using an established protocol including a FITC labeled anti-mouse lineage cocktail, APC labeled anti-mouse CD117 and APC-Cy7 labeled anti-mouse CD45 antibody (7). The number of ILC3 cells were similar in wildtype and miR-34a-/- mouse colons (Figure 2—figure supplement 1A). RT-qPCR showed similar IL-17A levels in wildtype and miR-34a-/- ILC3 cells (Figure 2—figure supplement 1B).

We measured IL23R expression in WT and miR-34a-/- CD4^+^ T cells (Figure 3—figure supplement 1).

Of additional concern, your team has tested for a role for stat3 and claim that IL17A activates stat3. Since IL17R's are known to signal through NF-κB, are these receptors also activating stat3 or is this a secondary effect of a downstream cytokine (for example IL-19/20/ IL-24) being produced? Again, it is difficult to make this link without the underlying mechanisms or at least a possible discussion of one. Related to this, what is the level of epithelial IL17RD expression in mir34aKO animals? Does treatment of infected mir-34a KO with anti-IL17A abrogate pathology and tumorigenesis?

Western blot showed that IL17 activates NF-κB in addition to STAT3 (Figure 4H). Treating colon organoids with an NF-κB inhibitor, BAY 11-7082, abrogated IL17-induced STAT3 phosphorylation and organoids growth (Figure 4H-J). Therefore, as the reviewer suggested, IL17 seems to activate STAT3 through NF-κB. We thank the reviewer for this suggestion.

Loss of miR-34a increased IL-17RD expression according to western blot (Figure 3—figure supplement 1).

We had anticipated the important question of anti-IL17A treatment and were in the middle of this experiment when we submitted our first version. After infection, we treated the mice with anti-IL17 neutralizing antibody in the first 2 months, stopped treatment in the next 4 months, and sacrificed the mice at the end of month 6 post infection. The anti-IL17 treatment suppressed tumorigenesis (Figure 6J-L).

In your Lgr5-GFP-CreER; mIR34a fl/fl mouse model, how are the mRNA or protein levels of proposed mIR34a downstream targets affected? CCL22, IL17 etc. (Figure 6A). Figure 6D is lacking the ko>ko and WT>WT BM chimera control. It is difficult to interpret the use of WT controls, which have not undergone irradiation (a treatment known to affect the intestinal compartment).

We performed western blot for IL6R, IL23R, CCL22 and IL17RD in the Lgr5-GFP-CreER; mIR34a fl/fl mouse model. The miR-34a conditional knockout did not affect IL6R and IL23R expression in CD4^+^ T cells, but increased CCL22 and IL17RD expression in the colon epithelium (Figure 3—figure supplement 1E-H).

We agree that control experiments are important and we should have thought of them. However, they took more than 6 months to complete. Following the feedback from the editor and reviewers, we discussed the limitation of this experiment and clarified that we are unable to make conclusions about radio resistant cell types without the control chimera.

Finally, reviewer 3 points out that the volcano plots shown in the supplementary figure should be complemented by including cumulative distribution (cdf) plots with logFC on the x axis comparing predicted miR34a targets to the rest of the transcriptome. These plots should show preferential derepression of the miR-34a targets (i.e. a shift to the right of the cdf plot compared to background). These are standard plots in the miRNA field and the reason for asking to include them has to do with the initial concerns raised regarding the absolute levels of miR-34a in these tissues. Demonstrating a signal attributable to miR-34a loss in the cdf plots would be conclusive evidence that the levels of miR-34a in these tissues are high enough to cause substantial target repression.

We thank the reviewer’s suggestion and followed previous examples (8, 9) to add the cumulative distribution plot (Figure 3 —figure supplement 4D).

1) Sato T, Stange DE, Ferrante M, et al. Long-term expansion of epithelial organoids from human colon, adenoma, adenocarcinoma, and Barrett's epithelium. Gastroenterology 2011;141:1762-72.

2) Sato T, van Es JH, Snippert HJ, et al. Paneth cells constitute the niche for Lgr5 stem cells in intestinal crypts. Nature 2010.

3) Weigmann B, Tubbe I, Seidel D, et al. Isolation and subsequent analysis of murine lamina propria mononuclear cells from colonic tissue. Nat Protoc 2007;2:2307-11.

4) Geem D, Medina-Contreras O, McBride M, et al. Specific microbiota-induced intestinal Th17 differentiation requires MHC class II but not GALT and mesenteric lymph nodes. J Immunol 2014;193:431-8.

5) Wang C, Kang SG, HogenEsch H, et al. Retinoic acid determines the precise tissue tropism of inflammatory Th17 cells in the intestine. J Immunol 2010;184:5519-26.

6) Artis D, Spits H. The biology of innate lymphoid cells. Nature 2015;517:293-301.

7) Gronke K, Kofoed-Nielsen M, Diefenbach A. Isolation and Flow Cytometry Analysis of Innate Lymphoid Cells from the Intestinal Lamina Propria. Methods Mol Biol 2017;1559:255-265.

8) Agarwal V, Bell GW, Nam JW, et al. Predicting effective microRNA target sites in mammalian mRNAs. *eLife* 2015;4.

9) Lewis BP, Burge CB, Bartel DP. Conserved seed pairing, often flanked by adenosines, indicates that thousands of human genes are microRNA targets. Cell 2005;120:15-20.

[Editors' note: further revisions were requested prior to acceptance, as described below.]

Reviewer #2:I have one comment that can be addressed textually.In Figure 2—figure supplement 1 the authors claim to evaluate ILC3 based on CD117+ CD45+ cells. This is in no way a specific straining strategy for ILC3 and the authors should revise their claims of ILC3s not being effected in their system based on this rather rudimentary analysis.

We thank reviewer 2 for his/her in-depth knowledge on ILC3. We have revised the relevant sentence in the manuscript text as follows:

“In the infected colon, miR-34a deletion did not significantly increase the number or IL-17 expression level of lineage(CD3e/Ly-6G/Ly-6C/CD11b/CD45R/B220/TER-119)-/CD117+/CD45+ cells, which contain a subset of ILC3 cells that may express IL-17 [Dong, 2008] (Figure 2—figure supplement 3A, B). Nevertheless, more markers will be needed to distinguish ILC3 and its subtypes specifically.”

We have also revised the legend for Figure 2—figure supplement 3A accordingly:

“FACS plots based on CD117 and CD45 markers performed on lineage- cell populations isolated from colons of *C. rodentium-*infected wild-type and miR-34a-/- mice.”